# Bond Strength and Fracture Toughness of Alkali Activated Self-Compacting Concrete Incorporating Metakaolin or Nanosilica

Radhwan Alzeebaree [1,2] 

1  Department of Highways and Bridges, Duhok Polytechnic University, Duhok 42001, Iraq; radhwan.alzeebaree@dpu.edu.krd; Tel.: +964-750-456-2178
2  Department of Civil Engineering, Nawroz University, Duhok 42001, Iraq

**Abstract:** This study aims to evaluate the effect of nanosilica (NS) and metakaolin (MK) as binder replacement materials on the fresh and hardened characteristics of slag (GGBS)-based alkali-activated self-compacting concretes (A-ASCC). Therefore, nine A-ASCC mixes, with and without metakaolin, were prepared, as well as mixes with and without NS incorporation. In the production of A-ASCC mixes, GGBS was used as a binder material. The fresh properties of A-ASCC were determined using the L-box, V-funnel, T50 value, and slump flow tests, while the hardened properties were examined using compressive strength, bonding strength (pullout test), fracture toughness, and flexural tensile strength tests. A relationship analysis was also conducted on the A-ASCC experimental data. The experimental results showed that NS and MK had a negative effect on the fresh properties of GGBS-based A-ASCC mixtures, whereas metakaolin had a greater influence. The addition of 1% and 2% NS, on the other hand, improved the mechanical performance of the A-ASCC specimens significantly. The use of more than 2% NS had a harmful effect on the mechanical properties of A-ASCC. A 5% replacement ratio of metakaolin improved the mechanical properties of A-ASCC. The use of metakaolin at ratios of more than 5% had a negative effect on the properties of A-ASCC.

**Keywords:** bond strength; fracture toughness; fresh properties; alkali-activated self-compacting concrete; nanosilica; and metakaolin

## 1. Introduction

Concrete is a composite material of different sized aggregates bound together by a cementitious binder material. Concrete is the most often used building material because of its durability, strength, and workability. Concrete has been man's most significant resource for survival and development, protecting billions of people. However, as cities become more densely populated, the overuse of concrete has begun to damage civilization due to its harmful environmental impact. Global attempts to produce environmentally friendly products have been unrelenting. Steel and concrete industries are large $CO_2$ emitters, and they are motivated to reduce $CO_2$ emissions by 2030 [1–3]. The energy cost of PC in concrete design is considerable, and studies indicate that for each kg of produced cement, $CO_2$ emissions range between 0.66–0.82 kg [4–6].

Therefore, it is essential to replace OPC with sustainable green materials, such as the use of alkali-activated concrete (AAC) produced using sustainable by-products, agricultural waste, and natural pozzolanic materials such as fly ash (FA), GGBS, and rice husk ash (RHA), and materials including silica and alumina components. The primary oxides found in the GGBS are calcium, silicon, aluminum, and magnesium oxides [3]. In the AAC technology, industrial waste/by-products materials rich in aluminosilicate work as cementitious binder materials [3]. Alkaline solutions such as potassium hydroxide, potassium silicate, sodium hydroxide, and sodium silicate activate the binder materials to produce AAC. When compared to conventional concretes, AAC produces less $CO_2$ and reduces land contamination caused by the disposal of by-products and industrial wastes. In

previous studies, fly ash was extensively utilized as the primary source of aluminosilicate. Many experiments have been conducted to replace fly ash with GGBS, with favorable results in the mechanical performance of concrete. However, few investigations have been conducted to study the characteristics of GGBS as a single binder material in alkali-activated self-compacting concrete (A-ASCC).

Self-compacting concrete, also known as fluid concrete, can expand through crowded reinforcement, filling all corners of formwork and compacting under its self-weight [7]. SCC exhibits high filling and passing ability, with excellent segregation resistance, or bleeding, without any vibration or compaction. Recently, the by-product and waste materials (AAB) were used as a binder to increase SCC's sustainability. The new product of SCCs with AAB is called alkali-activated self-compacting concrete (A-ASCC). Because A-ASCC varies from typical AAC in terms of workability and self-compaction, it is critical to understand the performance of A-ASCC and how it is affected. The favored choice of GGBS over other binder materials can be attributed to its higher calcium content and hydraulic reactivity, which allows it to cure under ambient environments, as compared to fly ash, which requires higher temperature curing. The alkali activation of GGBS varies from that of fly ash, and temperature is crucial to the process of activation. At ambient temperatures, the activation rate of FA particles is typically very low. When FAs are utilized as precursors, the alkali activation process often requires high temperatures. FA is alkali-activated in crystalline three-dimensional structures composed of alumina-silicate hydration [8–10]. The alkaline ions balance the negative charges in these hydrates. The behavior of normal AAC has already been well investigated [11]; however, further research into the fresh and hardened characteristics of A-ASCC is required to enhance this advanced type of concrete.

Recently, researchers focused on the development of AAC through the use of nano-materials [12–15]. The addition of NS to concrete has been investigated by various studies [16–18]. It was found that the addition of NS significantly improved the porosity and durability performance of AAC. This is owing to the eventual release of NS particles, which were initially suspended in the solution, in the reaction process at later stages, [19,20]. Naniz and Mazloom investigated the performance of SCC with various NS and W/B ratio proportions [21]. The mechanical characteristics of AAC were not improved by raising the nanosilica content by more than 2% by weight. This was due to the fact that the inclusion of nanosilica resulted in poor dispersion between the nanosilica and the geopolymer matrix, as well as in the creation of micro-voids, which weakens the AAC matrix [22,23]. The performance of concrete is significantly influenced by the bonding between the cement paste and the aggregate. Many studies found that utilizing NS increased the densified microstructure and C–S–H gel through nano-filler and anti-$Ca(OH)_2$-leaching, which improved the cement paste owing to the strong pozzolanic properties of nanosilica [24,25]. Moreover, the utilization of nanosilica improved the overall properties of hardened concrete by increasing and decreasing its total permeability [26–29].

Although there are many studies on alkali-activated concrete as a building material, the majority of studies have focused on FA-based AAC [30–32], GGBS [33,34], rice husk ash [35,36], or kaolin [37,38]. The studies on MK-based AAC have been done on mortar/paste to examine the interface between the alkaline solutions and the geopolymerization process [39–42], considering the effects of curing processes, such as temperature and/or time on the mechanical properties [43–46]. The effects of the ratio of the alkaline solution on the properties of MK-based AAC with the addition of polypropylene fiber at a ratio of 0.3, 0.5, and 1% were studied [47]. The fracture characteristics of MK-based AAC, mixed with a range of source materials, were investigated by Pires et al. [48]. Using a GGBS-MK mix, Xie et al. [49] evaluated the effect of GGBS and MK proportions, as well as recycled aggregate content, on fresh properties, compressive strength, Poisson's ratio, and toughness. The effect of concrete strength, heating rate, temperature level, and moisture content on spalling potential at high temperatures in AAC mixes produced from an MK/fly ash composition was also investigated [50].

Despite several studies on SCC, there have been few, if any, investigations on the hardened and fresh properties of A-ASCC. Therefore, the purpose of this study is to investigate how nanosilica and metakaolin affect the fresh, bond strength and mechanical characteristics of GGBS-based A-ASCC. As a result, two series of A-ASCC mixtures were produced: one including metakaolin, but no nanosilica, and the other with nanosilica, but no metakaolin. GGBS was replaced by weight with NS at a rate of 0%, 1%, 2%, 3%, and 4%, and replaced by weight with MK at a rate of 0%, 5%, 10%, 15%, and 20%, respectively. The workability and flowability of A-ASCC mixes were evaluated through the L-box ratio, V-funnel, slump flow, and T50 time tests. The compressive strength, flexural tensile strength, bonding strength, and fracture toughness of A-ASCC mixtures were determined while investigating their mechanical properties. All experimental test results were statistically evaluated, and a relationship analysis was shown to establish the effect of nanosilica and/or metakaolin on the properties of A-ASCC.

## 2. Experimental Procedure

### 2.1. Materials

Nanosilica (NS), metakaolin (MK), and fly ash were used to produce the A-ASCCs mixtures. NS, MK, and GGBS have specific surface areas of 15,000, 18,000, and 379 $m^2$/kg, respectively, and specific gravity values of 2.20, 2.54, and 2.27. The pH of NS in a 4% dispersion varied from 3.7 to 4.7, and the particle size was 14 nm. The GGBS was supplied by a steel reinforcement factory placed in the Mediterranean area of Turkey. The nanosilica was obtained from Norway. The metakaolin used in this study is a white powder with a whiteness value of 87, according to the Taylor-Lange measurement. The MK originated in the Czech Republic. Table 1 illustrates the chemical compositions and physical properties of the NS, MK, and GGBS. To achieve the desired workability and flowability, a commercially available Master Glenium 51-based superplasticizer (SP), with specific gravity of 1.07 kg/m³, was used. The crushed limestone fine and coarse aggregates were obtained from the same source. Table 2 shows the aggregate physical characteristics, as well as the sieve analysis. The alkali activator utilized in the production of A-ASCC was a mix of NaOH solutions and $Na_2SiO_3$. The component ratio of $Na_2SiO_3$ was found to be commercially available (water: 55.9%, $SiO_2$: 29.4%, and $Na_2O$: 13.7%, by mass). The NaOH purity ranged between 97 and 98%. In the current investigation, the molarity of NaOH was defined as 12 M, which has been recommended by numerous studies to achieve optimum mechanical and durability performance for A-ASCC [51].

**Table 1.** The physical and chemical characteristic of NS, FA, and MK.

| Component | CaO | $SiO_2$ | $Al_2O_3$ | $Fe_2O_3$ | MgO | $SO_3$ | $K_2O$ | $Na_2O$ | LOI | SG | BF ($m^2$/kg) |
|---|---|---|---|---|---|---|---|---|---|---|---|
| MK (%) | 1.287 | 50.995 | 42.631 | 2.114 | 0.127 | 0.439 | 0.337 | 0.284 | 1.640 | 2.54 | 18,000 |
| NS (%) | | 99.800 | | | | | | | <1.000 | 2.20 | 15,000 |
| GGBS (%) | 34.128 | 36.412 | 11.379 | 1.681 | 10.310 | 0.478 | 3.641 | 0.361 | 1.65 | 2.79 | 418 |

**Table 2.** The sieve analysis and physical properties of aggregates used in the production of A-ASCC.

| Sieve Size (mm) | 16 | 8 | 4 | 2 | 1 | 0.5 | 0.25 | Specific Gravity | Fineness Modules | Absorption |
|---|---|---|---|---|---|---|---|---|---|---|
| Coarse Aggregate | 100 | 31.0 | 1.5 | 0.75 | 0.25 | 0.25 | 0.65 | 2.72 | 5.65 | 2.4 |
| Fine Aggregate | 100 | 100 | 100 | 66.3 | 40.9 | 27.4 | 17.4 | 2.45 | 2.56 | 1.5 |

### 2.2. Mix Proportions

Two series of A-ASCC mixes were prepared using a constant 100% GGBS binder with 1, 2, 3, or 4% nanosilica, or 5, 10, 15, or 20% MK replacement ratio by weight of GGBS. The total binder content is 500 kg/m$^3$. Table 3 shows the quantity of each component of A-ASCCs mixes (weight/m$^3$ concrete). For mixture designations, NS represents nanosilica, the number next to NS (0–4) represents the nanosilica replacement ratio, and the number in front of MK (0–5–10–15–20) represents the quantity of MK utilized in the production of the A-ASCCs mixes. The mix designated (0MKNS0) in the 2nd row and 7th row of Table 3 are the same, for comparison purposes among the mixes, with and without NS or MK.

**Table 3.** Component of GGBS-based A-ASCCs mixes.

| Code of Mixture | Binder | Na$_2$SO$_3$ + NaOH | GGBS | NS | MK | Fine Agg. | Coarse Agg. | Molarity | SP | Extra Water |
|---|---|---|---|---|---|---|---|---|---|---|
| | kg/m$^3$ | kg/m$^3$ | kg/m$^3$ | kg/m$^3$ | kg/m$^3$ | kg/m$^3$ | kg/m$^3$ | | % | % |
| 0MKNS0 | 500 | 250 | 500 | 0 | 0 | 803.00 | 777.27 | 12 | 7 | 9 |
| 0MKNS1 | 500 | 250 | 495 | 5 | 0 | 802.34 | 776.63 | 12 | 7 | 9 |
| 0MKNS2 | 500 | 250 | 490 | 10 | 0 | 801.69 | 776.00 | 12 | 7 | 9 |
| 0MKNS3 | 500 | 250 | 485 | 15 | 0 | 801.03 | 775.36 | 12 | 7 | 9 |
| 0MKNS4 | 500 | 250 | 480 | 20 | 0 | 800.37 | 774.73 | 12 | 7 | 9 |
| 0MKNS0 | 500 | 250 | 500 | 0 | 0 | 803.00 | 777.27 | 12 | 7 | 9 |
| 5MKNS0 | 500 | 250 | 475 | 0 | 25 | 801.64 | 775.95 | 12 | 7 | 9 |
| 10MKNS0 | 500 | 250 | 450 | 0 | 50 | 800.27 | 774.62 | 12 | 7 | 9 |
| 15MKNS0 | 500 | 250 | 425 | 0 | 75 | 798.90 | 773.30 | 12 | 7 | 9 |
| 20MKNS0 | 500 | 250 | 400 | 0 | 100 | 797.54 | 771.98 | 12 | 7 | 9 |

Note: Mix #1 and Mix #6 are the same (control mix) for comparison of the results of the mixes with and without NS or MK.

The amount and kind of binder, the amount and ratio of alkaline and aggregates, as well as the maximum grain size (D$_{max}$), all have a substantial influence on the mechanical and fresh performance of A-ASCC specimens. According to prior research, the ratio of Na$_2$SiO$_3$/NaOH is suggested, for economic reasons, to be in the range of 1.5 to 2.5 [52]. As a result, in the current investigation, this ratio was designed as 2.5.

The mixing procedure was started by mixing the binder materials and the aggregates, in a dry condition, for 2.5 min. After 1 min, the alkali activator, additional water, and SP were slowly added to the dry mix and mixed for another 2.0 min. The mixture was then mixed for a further 3.0 min to ensure the uniformity and homogeneity of the A-ASCCs mixes.

### 2.3. The Fresh Properties Tests of A-ASCCs

The test apparatus used to achieve the fresh characteristics of A-ASCC indicated by L-box height ratio, V-funnel flow time, T50 slump flow duration, and slump flow diameter is shown in Figure 1. All fresh properties testing was performed in agreement with the guidelines issued by the EFNARC committee for the manufacturing of conventional SCC [53]. The slump flow test is a sensitive test for describing the flowability of a fresh concrete mix in free flow conditions. Therefore, the results of this test are proposed to be stated for all SCC mixes. T50 time is the time required for the concrete mix to flow to a set diameter of 50 cm on the flow slump table [53]. The T50 test provides details on bleeding, homogeneity, and segregation resistance of the concrete mix, which may be obtained via T50 time measurement and/or visual observations during the test. The EFNARC divides the average slump flow into three types; these classes indicate the SCC uses. Table 4 shows the usual application regions, in addition to the upper and lower limitations for these classes. The A-ASCC's viscosity is specified using the V-funnel flow duration and the T50

slump flow duration. These tests, however, cannot indicate the actual viscosity; instead, the flow rate in terms of viscosity is specified. The time assessed by the V-funnel test, which is the time elapsed for the mix flow via the V-funnel opening, is referred to as the time of V-funnel flow. The EFNARC viscosity classifications are illustrated in Table 4 [53]. The L-box test determines the flow capacity of fresh concrete through tight openings and restricted places, like semi-full reinforcement locations, without loss of homogeneity and segregation. The L-box ratio classifications are presented in Table 4.

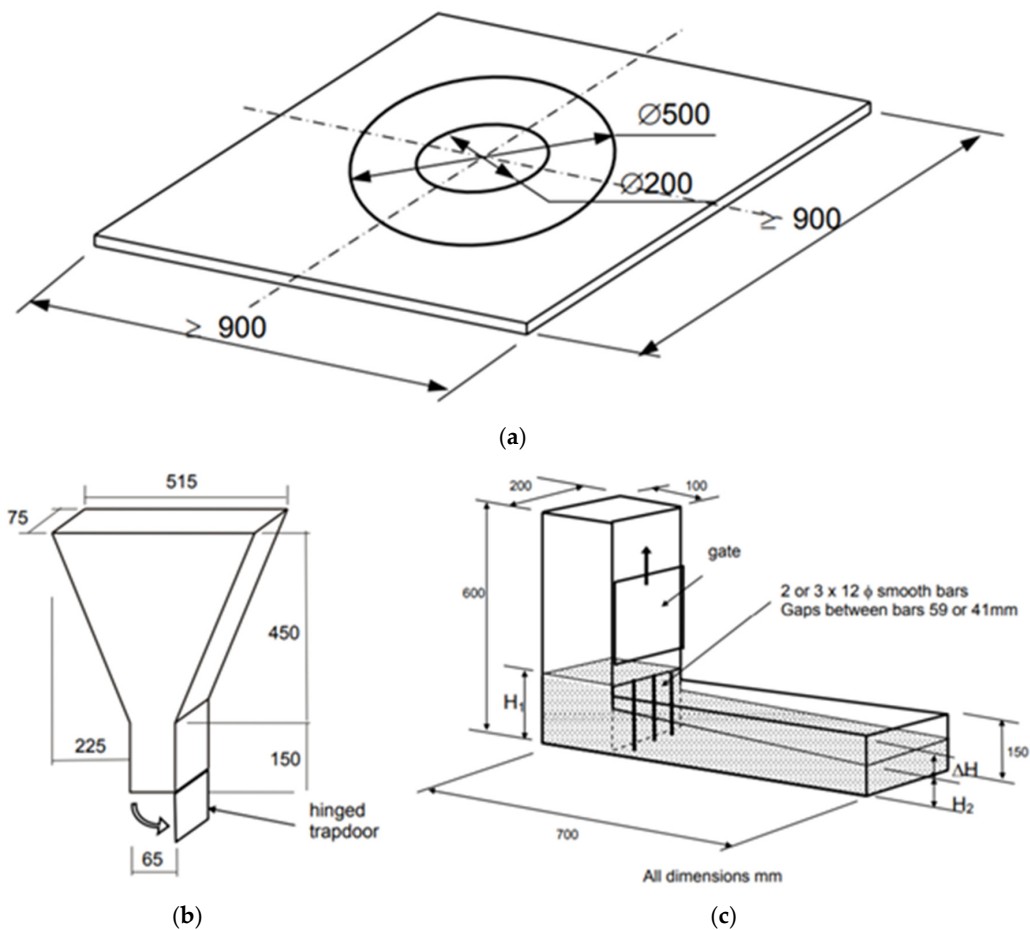

**Figure 1.** The test devices used to conduct testing on the fresh properties of A-ASCC mixes (**a**) slump flow test; (**b**) V-funnel flow test; (**c**) L-box passing ability test [54].

**Table 4.** The EFNARC specification and the classification of A-ASCC mix fresh properties.

| Classification of Slump Flow (SF) | | |
| --- | --- | --- |
| Class | SF diameter [cm] | |
| SF 1 | 55–65 | |
| SF 2 | 66–75 | |
| SF 3 | 76–85 | |
| **Classification of Viscosity** | | |
| Class | $T_{50}$ [s] | V-funnel [s] |
| VS1/VF1 | $\leq 2$ | $\leq 8$ |
| VS2/VF2 | >2 | 9 to 25 |
| **Classification of passing ability (PA)** | | |
| PA1 | $\geq 0.8$ with two rebar | |
| PA2 | $\geq 0.8$ with three rebar | |

*2.4. Curing Condition of the A-ASCC Specimens*

The A-ASCC specimens were covered with a protective sheet to keep the alkaline solution from evaporation. After the casting procedure, the specimens were placed in plastic bags to keep the alkaline solution from evaporating. Because the strength improvement was found to be limited after 48 h, the specimens and molds were put in an oven at 70 °C for 48 h to start the geopolymerization process [54,55]. The A-ASCC specimens were placed in the laboratory environment at 23 °C for 28 days after the oven curing process. For each experimental test, three typical specimens were prepared, and the average value of the corresponding investigative outcome was calculated.

*2.5. The Hardened Properties Tests for A-ASCC*

2.5.1. Compressive Strength

A-ASCC hardening tests were conducted to assess the influence of MK and NS on the mechanical performance of A-ASCC. Cubic specimens (100 × 100 × 100 mm) created using the ASTM C39 specification [56] were used to specify the compressive strength values of the A-ASCC specimens.

2.5.2. Flexural Tensile Strength

RILEM 50-FMC/198 Committee three-point flexural tensile strength tests were conducted on notched prismatic specimens having dimensions 100 × 100 × 500 mm [57]. The closed-loop displacement-controlled (Instron 5500R) machine was used to measure the flexural tensile strength. The deflection of the notched prismatic specimens was measured using a linear variable displacement transducer (LVDT). Notches with a width of 3 mm and a height of 40 mm were made (notch/depth: 0.4) at the specimens' bottom mid-point. Specimens were tested under displacement control at a rate of 0.02 mm/min. The 1st equation was utilized to compute the flexural tensile strength of A-ASCC specimens [58].

$$f_{flex} = \frac{3P_{\max}L}{2b(d-a)^2} \tag{1}$$

where $L$, $P_{\max}$, $b$, $d$, and $a$ represent the span length (mm), peak load (N), specimen width (mm), specimen depth (mm), and notch depth (mm), respectively. Figure 2 illustrates the three-point bending test setup, as well as the specimens that were tested.

The fracture energy ($G_f$) of the A-ASCC prismatic specimens was computed using the RILEM formula under three-point bending stresses, as illustrated below [57]:

$$G_f = \frac{(w_o + mg\delta_s)}{A_{lig}} \tag{2}$$

where $A_{lig}$, $\delta_s$, $g$, $m$ and $w_o$ are the area of the ligament (m$^2$), specific displacement (m), acceleration resulting from gravity (9.81 m/s$^2$), the mass of the beam (kg), and the area below the curve deducted by load-displacement relation (N-m), respectively.

Using the fourth equation below, the $K_{IC}$ (critical-stress-intensity factor) was also determined [59]:

$$K_{IC} = \frac{3P_{\max}l}{2bd^2}\sqrt{a_0}(1.93 - 3.07A + 14.53A^2 - 25.11A^3 + 25A^4) \tag{3}$$

where $l$, $P_{\max}$, $b$, $d$, $a_0$, and $A$ represent the span length, peak load, specimen width, specimen depth, notch depth, and notch depth to specimen depth, respectively.

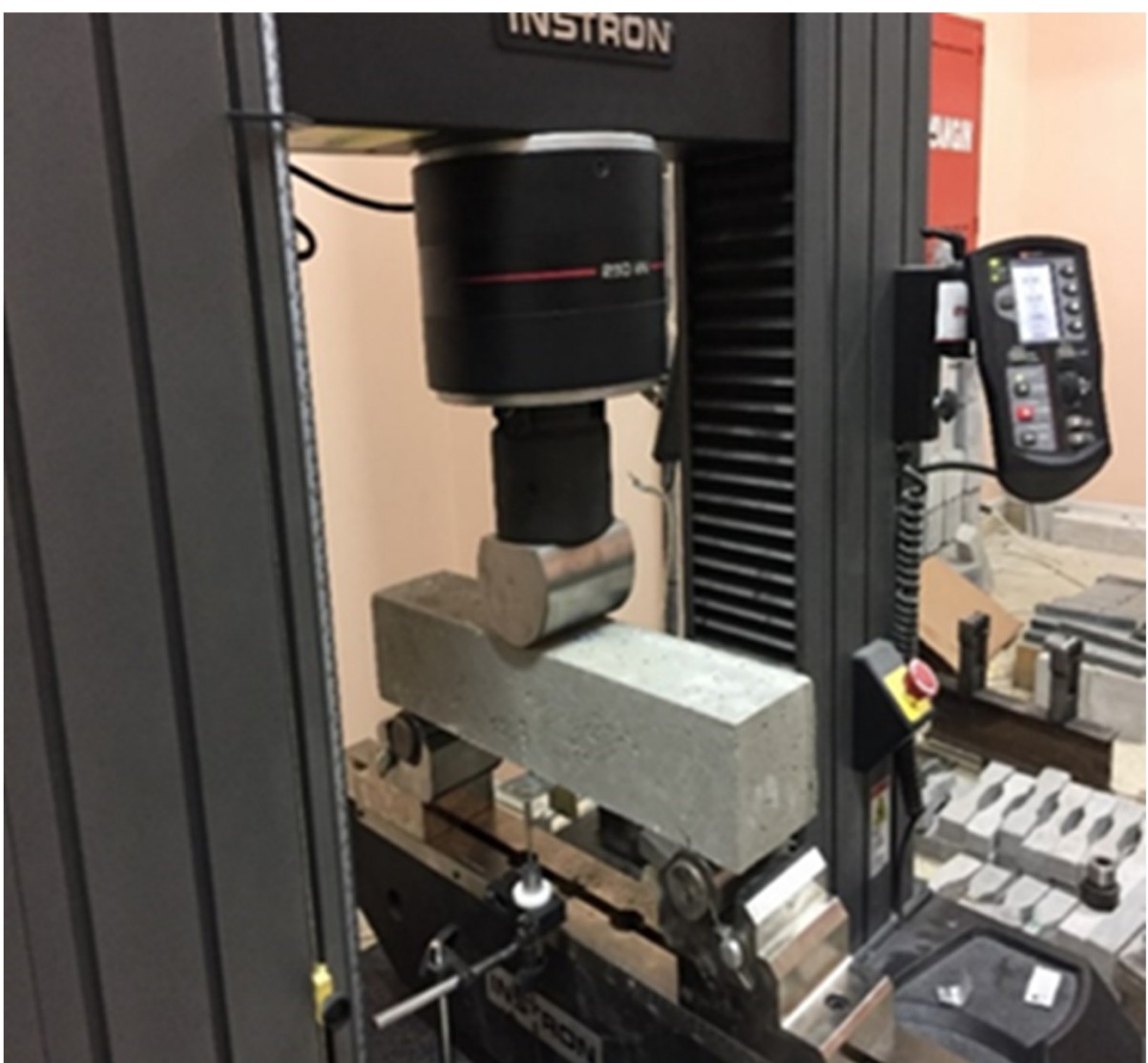

**Figure 2.** Test apparatus and specimens exposed to three-point bending loading.

### 2.5.3. Pullout Test and Bond Strength of A-ASCC

RILEM RC6 was conducted to measure the strength of the bond between steel reinforcing bar and concrete [60]. The upper surface of the pullout specimens was covered with gypsum to produce a flat surface for homogenous load distribution.

Figure 3 shows the pullout test setup, as well as the specimens tested under the pullout test. The bonding strength of A-ASCC is estimated using the 4th equation via the tensile force divided by the surface area of the embedded steel rebar inside the concrete:

$$\tau = \frac{F}{\pi \times d \times L} \tag{4}$$

where *d* is the diameter (mm), *F* is the failure tensile load (N), and *L* is the embedment length (mm) of the reinforcement steel bar. *L* and *d* in this investigation are 150 mm and 16 mm, respectively.

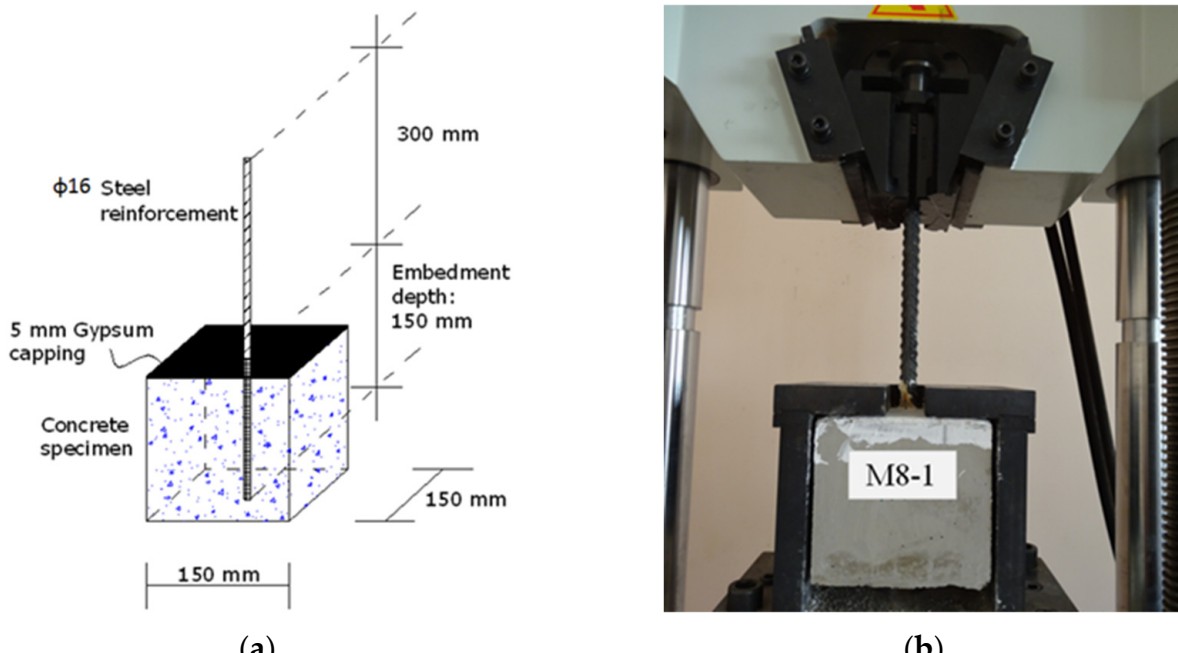

**(a)**                                                          **(b)**

**Figure 3.** The test setup and specimens used in the pullout test. (**a**) The dimensions and details of the A-ASCC specimen used for the bonding strength test; (**b**) A-ASCC specimen under the pullout test.

## 3. Result and Discussions

### 3.1. The Fresh Properties of A-ASCC

The fresh properties of the A-ASCC mix results, represented by flowability and passing ability tests, confirmed the EFNARC standard for SCC [53]. The slump flow results were noted to be within the range of EFNARC values, with the minimum value being 665 mm, which is more than the lowest slump flow of 550 mm stated in the EFNARC requirement. There was no segregation or bleeding in the A-ASCC mixtures. Furthermore, the slump flow values of A-ASCC matched those stated in the EN 12350-8 standard [61]. This standard requires a minimum slump flow value of 600 mm, which is less than the minimum values in the current investigation, as indicated in Table 5. The influence of MK and NS on the fresh properties, flowability, and passing ability of A-ASCC is investigated independently for each fresh test property, and the results are reported in the sections below.

**Table 5.** The A-ASCC fresh test results.

| Mixture | S-FLOW | L-BOX | V-FUNNEL | T-50 |
|---------|--------|-------|----------|------|
| 0MKNS0 | 703.00 | 0.95 | 15.03 | 3.22 |
| 0MKNS1 | 712.00 | 0.96 | 14.42 | 3.05 |
| 0MKNS2 | 719.00 | 0.97 | 13.96 | 2.93 |
| 0MKNS3 | 727.00 | 1.00 | 13.28 | 2.85 |
| 0MKNS4 | 731.00 | 1.00 | 12.63 | 2.64 |
| 5MKNS0 | 689.00 | 0.91 | 16.85 | 3.57 |
| 10MKNS0 | 673.00 | 0.87 | 17.67 | 3.81 |
| 15MKNS0 | 662.00 | 0.81 | 19.80 | 4.11 |
| 20MKNS0 | 651.00 | 0.79 | 24.50 | 4.35 |

On the other hand, the setting time and the hardening for all mixes were substantially decreased with the replacement of MK and increased with addition of NS. Researchers studied the use of alkali-activated concrete in ambient environments [12,22]. It can be noted that the replacement of MK increases the probability of the use of A-ASCC under an ambient environment. However, the addition of NS might be use to eliminate the shrinkage behavior of GGBS based A-ASCC. Researchers found that the surface charge of a GGBS

particle influences the early initial AAM mix setting time [62]. The early setting times of the GGBS-based A-ASCC containing 1%, 2%, 3% and 4% NS were 1 h, 1.25 h, 1.5 h, and 1.5 h, respectively. The ultimate setting times for these mixes were 6 h, 7 h, 8 h, and 8 h, respectively. Therefore, NS might be used to control the shrinkage of GGBS-based A-ASCC, as stated by researchers [63]. Blending the GGBS with MK resulted in even greater reductions in setting time. The initial setting times of the GGBS-based A-ASCC containing 5%, 10%, 15%, and 20% MK were 0.5 h, 0.5 h, 0.75 h, and 1.0 h, respectively. The ultimate setting times for these mixes were 4 h, 4 h, 3 h, and 2 h, respectively.

### 3.1.1. Slump Flow Test

The influences of MK and NS on the slump flow values of A-ASCC are presented in Figure 4. The presence of NS increased the slump flow of the A-ASCC mixes. The maximum slump flow value (731 mm) was deducted from the mix including GGBS with 4% NS and without MK (0MKNS4). However, the addition of MK alone, without NS, decreased the flow values of the A-ASCC mixes from 731 mm without MK to 689 mm with 5% MK, 673 mm with 10% MK, 662 mm with 15% MK, and 651 mm with 20% MK, respectively. Moreover, the addition of NS increased the flow values of A-ASCC, and the increment ratio was 1.3%, 2.3%, 3.4%, and 4% for the mixtures including 1% NS, 2% NS, 3% NS, and 4% NS, respectively, whereas the influence of MK was greater than NS. Furthermore, the addition of additional NS and/or MK to the A-ASCC mixture increased resistance to bleeding and segregation, and NS- or MK-containing mixtures were shown to be more cohesive than non-NS or MK-containing mixtures. The favorable effect of nanosilica on the fresh characteristics of A-ASCC and its ability to minimize the initial or final setting time may make it a viable solution to the shrinkage problem, which is an ongoing problem for GGBS-based AAC.

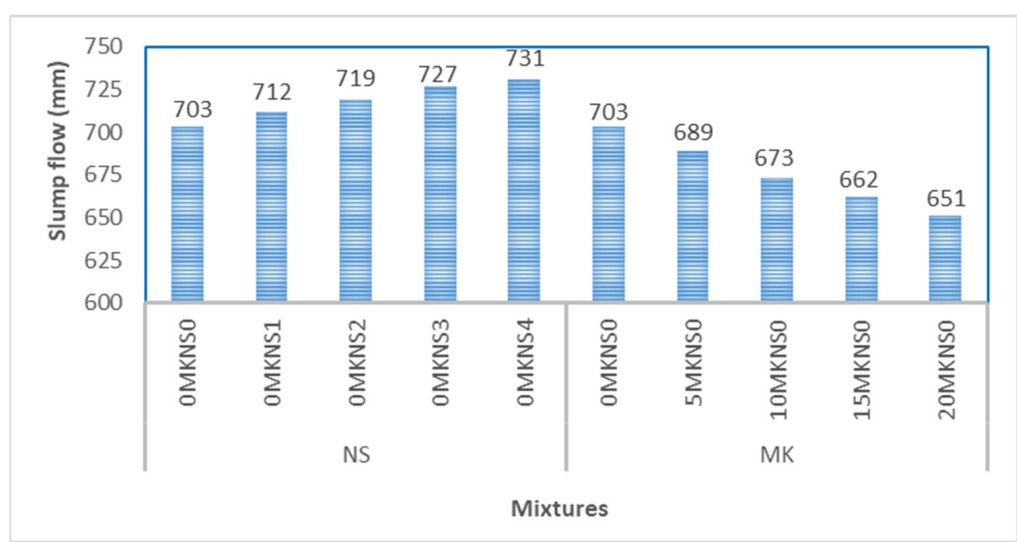

**Figure 4.** The slump flow values of A-ASCC mixtures vs. the MK and NS replacement ratios.

Furthermore, it was shown that the results of the slump flow testing for all A-ASCC mixes were in the range of SF2 in the classification of the EFNARC. According to the EFNARC specifications, this class is compatible with several structural applications (slabs, beams, and columns) [53]. Meanwhile, the A-ASCC mixes in the current study are suitable for structural applications, such as beams, slabs, and columns. Previous research on conventional SCC found that using NS reduced the slump flow value when compared to FA-mix (control mixes). This is because NS, with high surface area particles, absorbed some of the mixing water and decreased the flow values. Water molecules are drawn to NS particles because of their high reactivity and higher surface area. Consequently, the free water required to improve the flowability of the mix is reduced [64]. However, the use of 2% NS with 50% FA and 50 GGBSs was shown to exhibit rare or no effects on the fresh

properties of A-ASCC [22]. Researchers investigated the influence of MK on the fresh and hardened characteristics of conventional SCC and revealed that SCC with MK can achieve satisfactory workability [65].

Furthermore, the setting time and/or hardening time for all mixes were decreased with the increase in the replacement ratios of MK, while the addition of NS shows a negative effect on the setting time of A-ASCC. Researchers studied the use of alkali-activated concrete in ambient environments [12,22]. The addition of MK increases the probability of the use of A-ASCC under an ambient environment.

### 3.1.2. T50 Time Test

The combined effect of NS and MK on the T50 times is presented in Figure 5. T50 represents the time measured until the fresh A-ASCC reaches 500 mm. T50 durations of the mixes without NS and including MK, with replacement ratio of 0%, 5%, 10%, 15%, and 20%, were evaluated as 3.22 s, 3.57 s, 3.81 s, 4.11 s, and 4.35 s, respectively, and 3.22 s, 3.05 s, 2.93 s, 2.85, and 2.64 s, respectively, for the mixes including 0%, 1%, 2%, 3%, and 4% NS. The increment in the values of T50 due to the addition of MK may be due to the MK's high surface area and the high absorption properties of MK particles. However, the reduction of T50 times due to the addition of NS may be related to the absorption properties of GGBS particles being much greater than that of the NS particles. Moreover, the T50 duration in the current study was less than 6 s, which is compatible with the EN 12350-8 standard [61]. Meanwhile, According to the EN 12350-8 standard [61] and EFNARC specifications [53], all T50 values in the current study were considered acceptable. In addition, the results of the T50 duration tests deducted that the flowability of A-ASCC was negatively affected by the amount of MK and/or NS; the lower T50 duration was detected in the mixes containing GGBS without MK and NS.

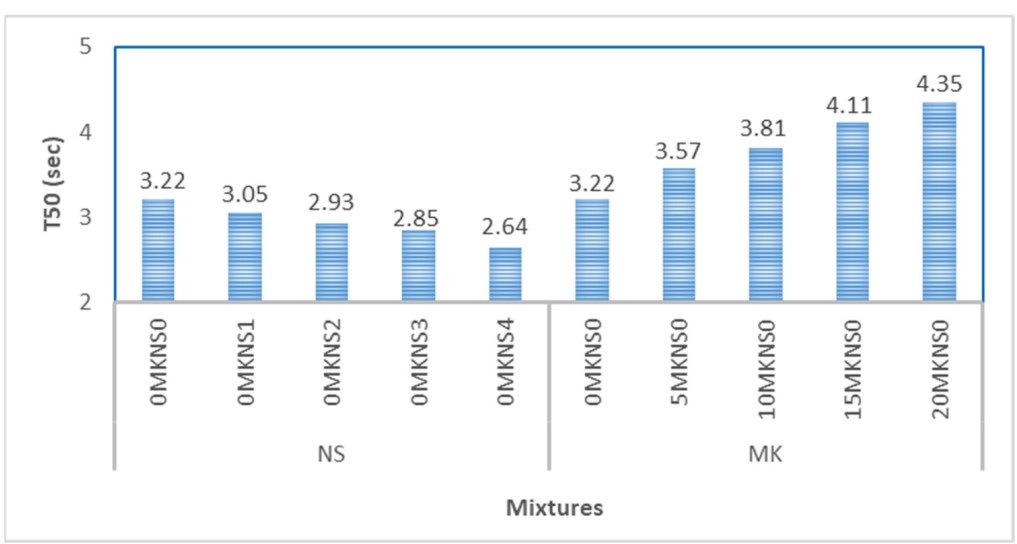

**Figure 5.** The T50 duration of A-ASCC mixes vs. MK and NS replacement ratios.

### 3.1.3. V-Funnel Flow Time of A-ASCC Mixes

The reflection and viscosity for the flowability of A-ASCC are represented by the V-funnel flow duration. Similar to the T50 duration and slump flow, the discharge time from the V-funnel opening showed the best relationship with the replacement ratio of NS. The V-funnel flow durations of the mixes without NS including MK were evaluated as 15.03 s, 16.85 s, 17.68 s, 19.80 s, and 24.50 s for the mixes including 0%, 5%, 10%, 15%, and 20% MK, respectively. However, the V-funnel flow durations for the mixes incorporating NS without MK were 15.03 s (0% NS), 14.42 s (1% NS), 13.96 s (2% NS), 13.28 s (3% NS), and 12.63 s (4% NS). As a result, the influence of MK on the flow time was found to be more than the influence of NS. Similarly, the minimum and maximum flow times were detected for the mix including 4% NS and the mix including 20% MK, respectively, as

indicated in Figure 6a. Furthermore, Table 5 and Figure 6b show the viscosity classes depending on the T50 slump flow duration values and the V-funnel time stated by the EFNARC specifications [53]. On the other hand, the amount of MK can significantly affect the mix flow time; Figure 6 indicated that the flowability times increased with an increase in the amount of MK replacement ratios and decreased with the addition of NS.

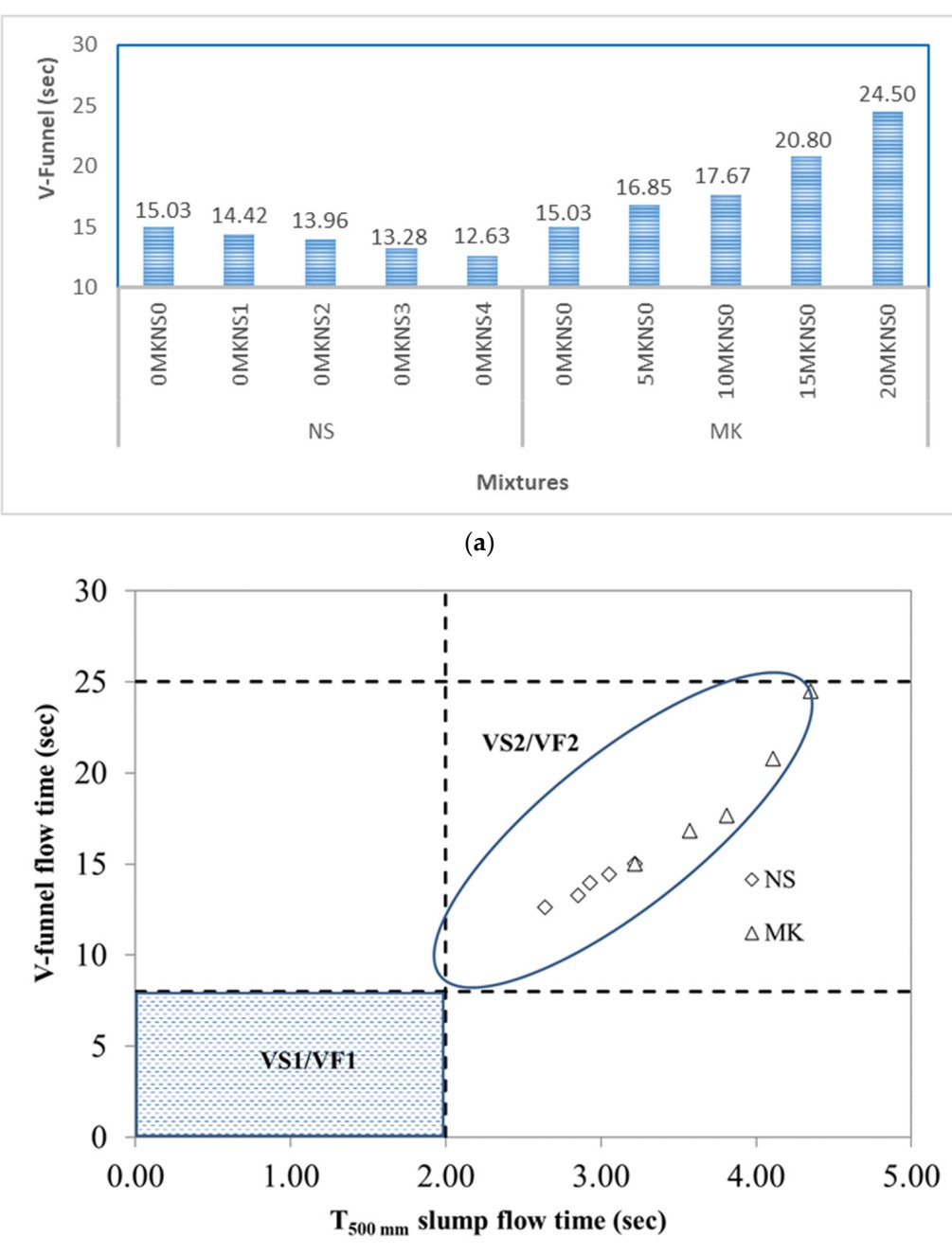

**Figure 6.** (**a**) The V-funnel duration of A-ASCC mixes vs. the NS and MK replacement ratio; (**b**) V-funnel time vs. T50 flow time relationships.

Based on the T50 duration and V-funnel discharge time measurements, all A-ASCC mixtures in the current investigation were classified as VS2/VF2. According to the EFNARC standard, the VS1/VF1 viscosity classification indicates a sufficient flow, even for high reinforcing structures. This class, however, has insufficient resistance to segregation and bleeding, while the VS2/VF2 class has minimal formwork pressure and excellent resistance

to bleeding and segregation. As a result, the surface finish property of the VS2/VF2 class is insufficient, and it may degrade as a result of the cessation of the flow of the A-ASCC mixes. According to the V-funnel results, the presence of NS enhanced discharge flow duration and improved bleeding and segregation resistance. Furthermore, according to the EN 12350-9 standard [66], the flow time of the V-funnel of any A-ASCC mix should be less than 15 s to display excellent filling ability. However, except for the mixes containing NS, the flow duration of the mixes with and without MK was more than 15 s (the normal limit value); the mixtures that include NS exhibit a flow duration of fewer than 15 s. Moreover, the flow duration of all A-ASCC mixes including MK was greater than the values of the mixes containing NS. Furthermore, higher concentrations of NS and MK enhanced the resistance of A-ASCC mixtures to bleeding and segregation, and the mixes containing MK and/or NS were shown to be more cohesive than the control mix (without MK and NS). The results revealed that, as the NS ratios increased, so did the fresh state properties of the A-ASCC mixtures. To produce GGBS-based A-ASCC mixes with superior flowability and passing ability, the proportion of both MK and NS should be controlled to obtain excellent fresh state performance.

### 3.1.4. The L-Box Height Ratio of A-ASCC Mixtures

The L-box test measures the flowability of A-ASCC mixes through a narrow open channel with three bars (411 mm) by calculating the ratio of H2/H1 (the height of A-ASCC in the horizontal part/the height of A-ASCC in the vertical part) after the mix has fully flowed. The EN12350-10 standard and EFNARC specification stated that the passing ability ratio (H2/H1) via the L-box test should be more than or equal to 0.8 to meet the requirements for the passing ability of SCC mixtures [67]. Figure 7 represents the results for passing ability using the L-box test in the current study. As seen from the figure, all A-ASCC mixes exhibit an acceptable passing ability (passing ability ≥ 0.8). The mixes without MK with and without NS have the highest passing ability (1.0). However, as expected, the passing ability of A-ASCC decreased with the addition of MK; the passing ability for the fibrous mixes with MK were 0.95, 0.96, 0.97, 1.0, and 1.0 for the mixes including 0%, 1%, 2%, 3%, and 4% NS, respectively. In addition, it was indicated that the incorporation of MK decreased the passing ability values for A-ASCC mixes. It should be noted that the passing ability of A-ASCC was reduced by increasing the amount of MK (from 5% to 20%). Furthermore, it was noted that the replacement of NS increased the passing ability of GGBS-based A-ASCC mixes.

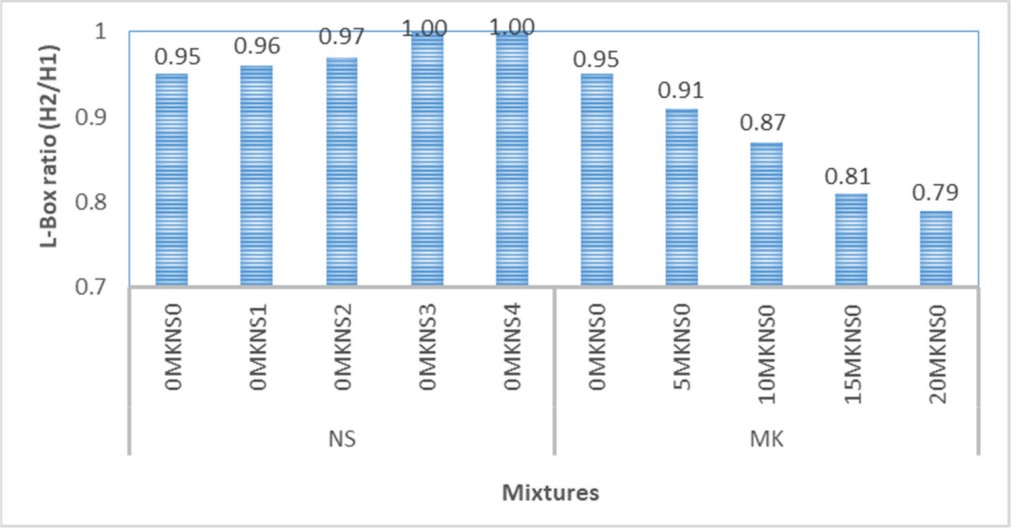

**Figure 7.** The effect of NS replacement level and MK content on the L-box height ratio.

On the other hand, it was indicated that all mixes (except 20% MK) in the current study satisfied the passing ability and flowability criteria requirements, according to TS EN 12350 standards and EFNARC specifications, even for the presence of the highest volume fraction (1%) for both the addition of NS and/or MK. Furthermore, it was noticed that the incorporation of NS and MK improved the fresh quality of GGBS-based A-ASCC mixes by improving the resistance to bleeding and segregation and enhancing the finishing surface and uniformity. Therefore, it is recommended to use NS and MK to obtain better fresh properties (the passing ability and flowability) of A-ASCC.

### 3.2. Hardened Performance of A-ASCC

#### 3.2.1. Compressive Strength

Table 6 and Figure 8 illustrate the values of compressive strength for A-ASCC specimens in the current study. The results showed that incorporating NS had no substantial influence on the compressive strength results of the A-ASCC specimen. Therefore, the compressive strength of the A-ASCC specimen containing NS was close to the corresponding compressive strength of the specimen without NS; the compressive strength of GGBS-based A-ASCC was slightly improved by the addition of 1% of NS and reduced with an increase in the replacement of NS. These results may be owing to a severe self-dehydration for the specimens under the presence of NS as a result of unreacted partial NS particles, which promotes crack development (weakening the microstructure) and hence, the deterioration of compressive strength [13,24]. The improvement ratio was 1.6% for the specimens including 1% NS, whereas the reduction amounts were 0.15%, 3.37%, and 4.54% for the specimens including 2%, 3%, and 4% NS, respectively. The loss in compressive strength was associated with the increase in the ratio of unreacted silica in the mixture, which increased the ratio of silicate-to-aluminate. As a result, the optimal NS ratio for the GGBS-based A-ASCC is 2%. Furthermore, the inclusion of 5% MK was shown to improve compressive strength substantially. The compressive strength enhancement was greater than 6.6%. However, the use of more than 5% MK exhibits a negative effect on the compressive strength of GGBS-based A-ASCC. The reduction amounts were 4.9%, 12.7%, and 15.1% for the specimens including 10%, 15%, and 20% MK, respectively. Previous studies have shown that the incorporation of MK improved the compressive strength of conventional concrete [68,69].

On the other hand, it was noted that the compressive strength of A-ASCC was enhanced with an increase in the time (from 7 days to 28 days). The compressive strength improvement for the specimens that include MK was much more than for the specimens without MK. However, the improvement of the compressive strength for the specimens that include NS was close to the improvement values for the control specimens (specimens without NS and MK)

**Table 6.** Hardened properties of A-ASCC mixtures.

| Mixtures | MK% | NS% | Compressive Strength (MPa) | Bond Strength (MPa) | Fracture Energy (N/m) | KIC (MPa-mm$^{1/2}$) | Net Flexural Strength (MPa) | Displacement (mm) |
|---|---|---|---|---|---|---|---|---|
| 0MKNS0 | 0% | 0% | 77.54 | 11.22 | 126.27 | 25.47 | 5.38 | 0.43 |
| 0MKNS1 | 0% | 1% | 78.81 | 11.23 | 126.76 | 25.41 | 5.37 | 0.44 |
| 0MKNS2 | 0% | 2% | 77.42 | 12.35 | 127.59 | 25.42 | 5.37 | 0.46 |
| 0MKNS3 | 0% | 3% | 76.19 | 11.09 | 126.11 | 25.03 | 5.28 | 0.49 |
| 0MKNS4 | 0% | 4% | 73.9 | 10.64 | 124.14 | 23.80 | 5.02 | 0.53 |
| 0MKNS0 | 0% | 0% | 77.54 | 11.22 | 126.27 | 25.47 | 5.38 | 0.43 |
| 5MKNS0 | 5% | 0% | 82.66 | 11.26 | 127.61 | 27.15 | 5.73 | 0.50 |
| 10MKNS0 | 10% | 0% | 73.72 | 10.91 | 125.44 | 24.21 | 5.11 | 0.45 |
| 15MKNS0 | 15% | 0% | 68.91 | 10.37 | 124.15 | 22.63 | 4.78 | 0.48 |
| 20MKNS0 | 20% | 0% | 65.68 | 9.74 | 123.18 | 20.63 | 4.35 | 0.51 |

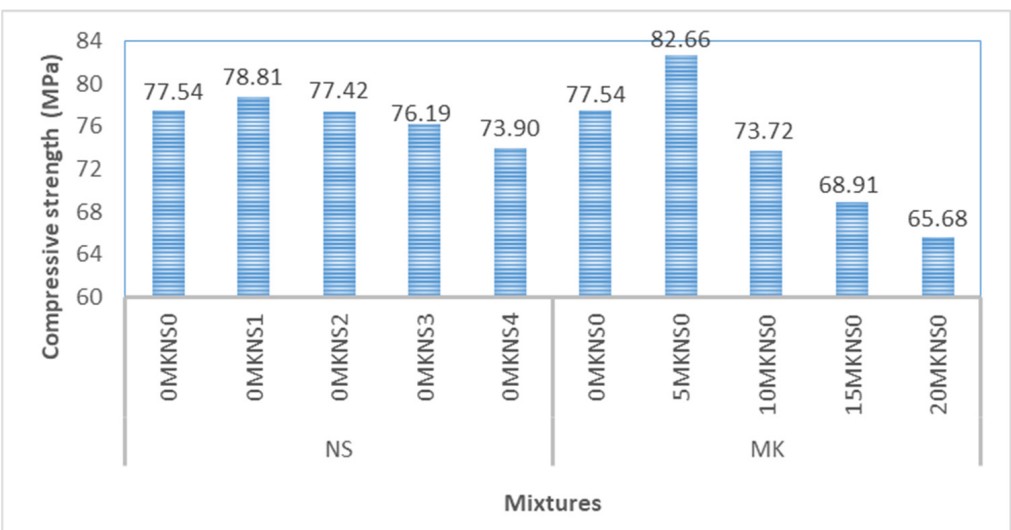

**Figure 8.** The compressive strength of the A-ASCC vs. NS and MK replacement ratios.

Saini and Vattipalli [70] studied the A-ASCC performance with the inclusion of NS. According to their findings, including 2% NS caused in a considerable development of compressive strength. Researchers showed that the NS densified the matrix structure of tetrahedral aluminosilicate, and therefore the addition of NS to A-ASCC mixtures enhanced their strength [71]. Nanomaterials have been shown to improve the behavior of cementitious composites [72,73]. However, researchers found that the influence of NS on the compressive strength of AAC and/or A-ASCC was very low compared to the tensile and bond strength behavior, regardless of the type of binder materials and curing environment used [22,24,74]. It can be concluded that the improvement ratio of the compressive strength for the specimens including NS depends considerably on the type and amounts of binder materials used. High-strength concrete has been investigated by Zareei et al. [75] using 2% NS and GGBS aggregate. It was demonstrated that adding 2% NS by weight of binder materials was the best proportion for completing the geopolimerization process, working as a cavity filler; however, this was highly dependent on the base materials utilized [24,63].

### 3.2.2. Bond Strength

Figure 9 shows the difference in bond strength caused by the addition of NS and MK calculated using the fourth equation. The results showed that incorporating NS and/or MK had no substantial influence on the bond strength results of the GGBS-based A-ASCC specimens. Therefore, the bond strength of the A-ASCC specimens containing NS and/or MK was close to the corresponding bond strength of the specimen without NS; the bond strength of GGBS-based A-ASCC was slightly improved by the addition of 1% of NS, 2% NS, and 5% MK, and reduced with an increase in the replacement of NS and/or MK. These results may be owing to a severe self-dehydration with the specimens' presence of NS as a result of unreacted partial NS particles, which promotes crack development (weakening the microstructure) and hence, the deterioration of bond strength [13,24]. The maximum bond strength improvement was reported to be 1% for specimens containing 2% NS when compared to control A-ASCC specimens. The favorable impact of NS on bond strength was found to be larger than the effect of MK. The improvement ratios were 0.7%, and 1.0% for the specimens that included 1%, and 2% NS, respectively, whereas the bond strength was improved slightly by the addition of 5% MK and negatively affected by the replacement ratio increment of MK (more than 5% MK). The reduction values of bond strength were 1.14%, and 5.2% for the specimens including 3% and 4% NS, respectively, and 4.0%, 7.5%, and 13.1% for the specimens including 10%, 15%, and 20% MK, respectively.

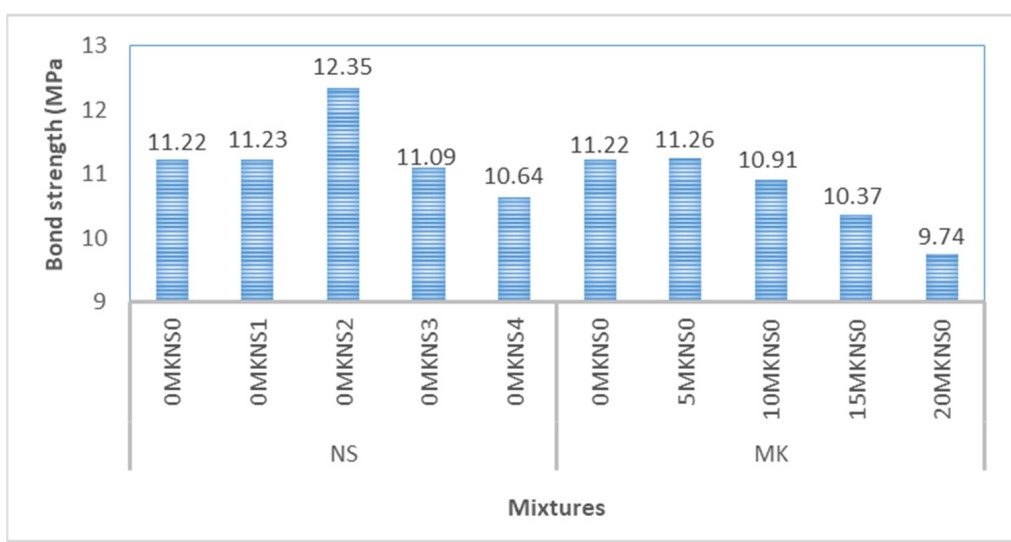

**Figure 9.** The combined influence of MK and NS replacement on bond strength.

Metakaolin-based AAM has demonstrated better bond strength when compared to comparable repair products on the market [76]. The alkaline particles are dissolved in a highly alkaline solution, resulting in amorphous polymeric Si–O–Al–O bonds. The geopolymer forms quickly and is a good contender for early strength applications [77,78]. In the current study, the bond strength of A-ASCC was significantly improved with the replacement of MK and NS, as shown in Figure 10. However, the improvement achieved by the NS replacement ratio was more than the improvement achieved by MK replacement.

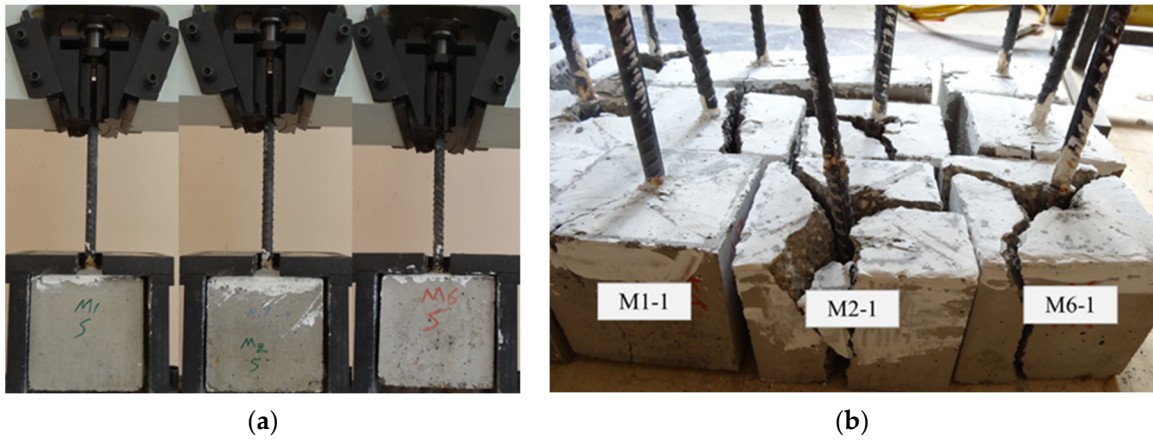

(**a**)                                                    (**b**)

**Figure 10.** Typical bond failure patterns of A-ASCCs: (**a**) without NS and/or MK, and (**b**) with NS and/or MK. (**a**) A-ASCC specimen before pullout test; (**b**) A-ASCC specimen after pullout test.

### 3.2.3. Net Flexural Tensile Strength

The load-displacement curves of A-ASCC specimens under three-point bending stress calculated using the 1st equation are presented in Figure 11. In general, all A-ASCC curves displayed a linear increasing trend until strain-softening behavior and initial cracking for the specimens was seen. The load-displacement curves of A-ASCC specimens exhibited comparable behavior, and flexural strength values were obtained with and without MK and/or NS (up to 2%). As illustrated in Figure 12, when MK and NS were added to the GGBS-based A-ASCC, the results showed that incorporating NS and/or MK had no substantial influence on the flexural tensile strength of the A-ASCC specimen. Therefore, the flexural tensile strength of the A-ASCC specimen containing NS and MK was close to the corresponding compressive strength of the specimen without NS and/or MK; the compressive strength of GGBS-based A-ASCC was slightly improved by the addition of

5% of MK and reduced with an increase in the replacement of MK and/or NS. These results may be owing to a severe self-dehydration for the specimens' presence of NS and/or MK as a result of unreacted partial NS and/or MK particles, which promotes crack development (weakening the microstructure) and hence, the deterioration of compressive strength [13,24]. Furthermore, specimens with higher NS (up to 2%) and/or MK (up to 5%) showed greater residual flexural strengths and lower load relaxations than specimens with lower NS and/or MK. This might be due to the increased bonding strength between the NS and/or MK particles and the matrix as a result of the NS and/or MK addition. The presence of MK, on the other hand, resulted in the greatest bond strength and maximum flexural strength for specimens containing 5% MK. Guneyisi et al. [69] investigated the effect of MK and steel fiber on the properties of conventional concrete; however, the effect of steel fiber was shown to be dominant on the bonding strength and flexural tensile strength. The combination of these two elements (steel fiber and MK) significantly improved the mechanical properties of the specimens. Previous research also evaluated the fracture energy, bending strength, and bond/or bond strength of A-ASCC with and without NS and steel fiber; the NS considerably enhanced the bending and bond strength of A-ASCC, and the enhancement increased with the combination of NS and steel fiber [12,22].

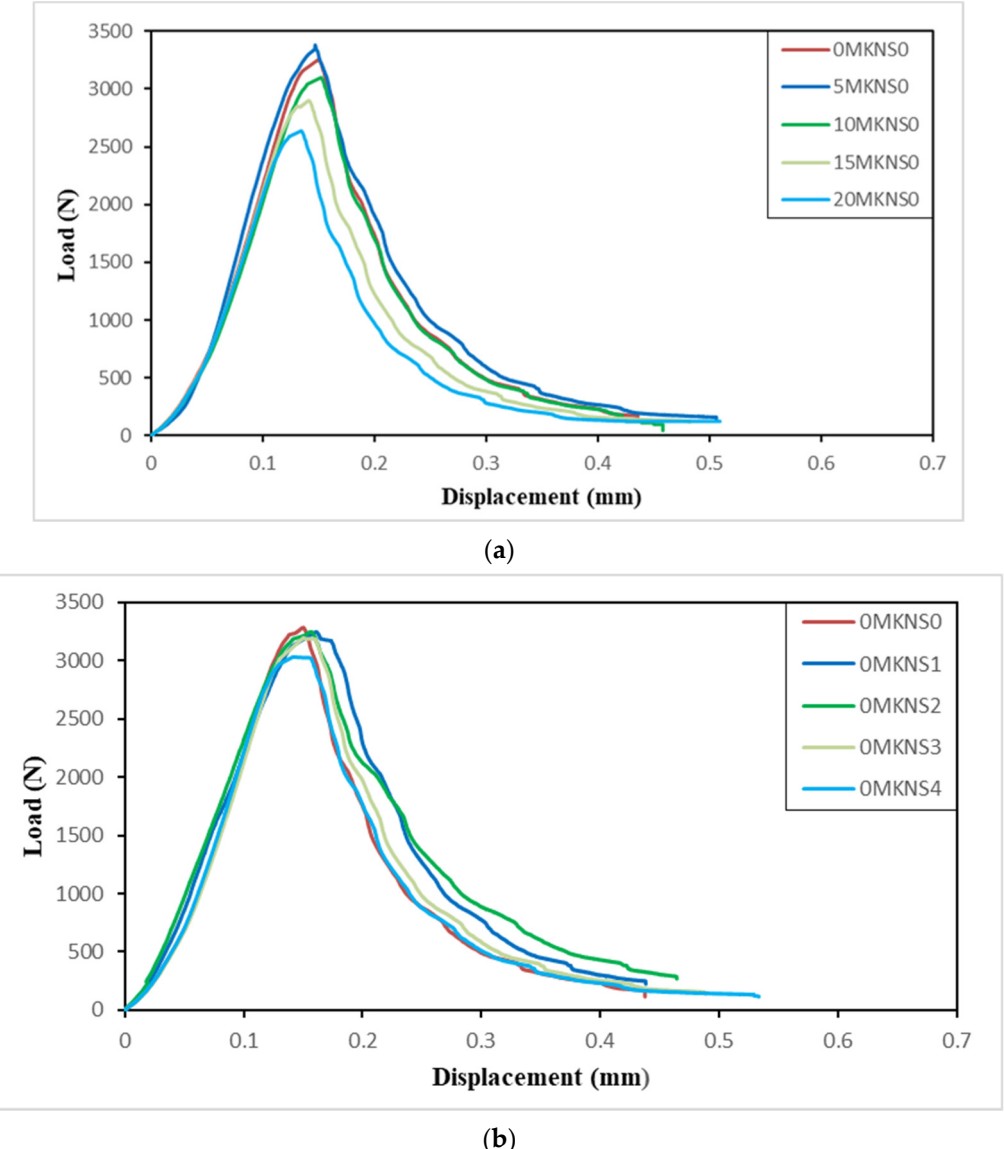

**Figure 11.** The load displacement of A-ASCC vs. MK and NS replacement ratios; (**a**) effect of MK on A-ASCC mixes; (**b**) effect of NS on A-ASCC mixes.

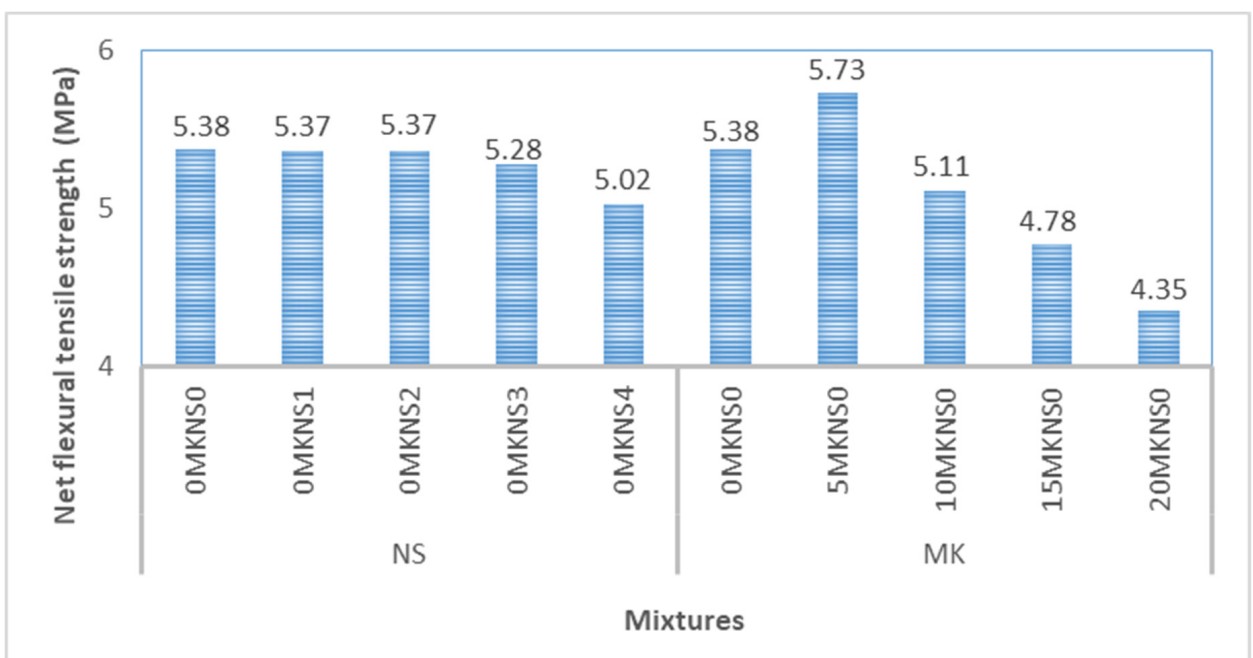

**Figure 12.** The flexural strength vs. MK and NS replacement ratios.

### 3.2.4. Fracture Performance

Figure 13 shows the fracture energy results from measuring the area under the load-displacement curve using the 2nd equation. For the control specimens, similar fracture energy findings were found (without MK and NS). When the influence of MK and NS on fracture energy was investigated, the fracture energy enhanced by increasing the amount of MK (up to 5%) and/or NS (up to 2%), but decreased with an increase in the ratio of NS (3% and 4%) and/or MK (more than 5%). The negative effect of NS might be attributed to an incomplete NS reactive and a lack of calcium to reactivate the particles of NS, while the unfavorable influence of MK on the fresh characteristics may be the primary influence on A-ASCC fracture toughness and mechanical properties.

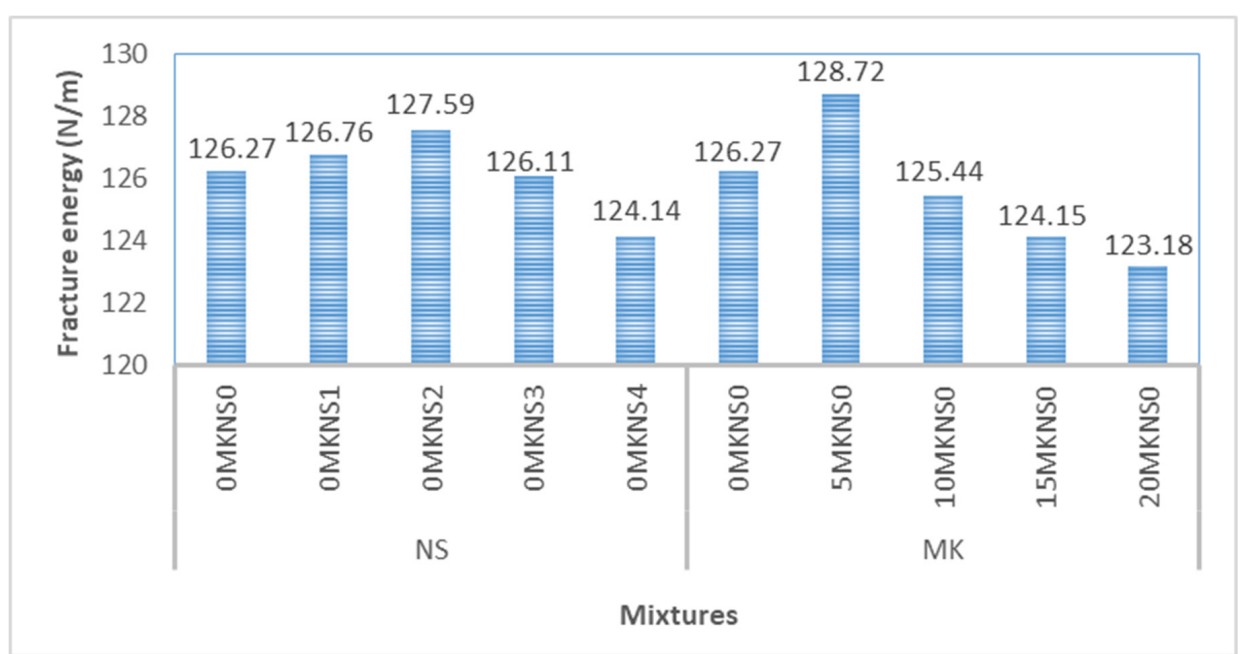

**Figure 13.** The fracture energy of A-ASCC vs. MK and/or NS replacement ratios.

The effect of MK and NS was attributed to increased bonding strength, which circumvents fractures and allows for higher displacement in the A-ASCC specimens. Meanwhile, MK and NS may strengthen bonds and increase matrix component adhesion. Another reason for the higher fracture energy values might be an increase in the compressive strength of the specimens, as seen in Figure 13. Previous studies investigated the fracture energy of heat-cured GGBS-based AAC and discovered that it increased as compressive strength increased [79].

The 3rd equation was used to measure the KIC (critical stress intensity factor) of A-ASCC specimens after 28 days, and the outcomes are given in Figure 14. The KIC indicates how much stress is necessary to propagate the fracture. The KIC values for A-ASCC specimens with and without MK and/or NS were similar. As seen in Figure 14, when the MK and NS content was increased (up to 5% MK and 2% NS), the KIC values increased as well. When 5% MK and 2% NS were used, more stress was required to form the existing cracks in the specimens. The KIC values were decreased as the NS and MK were increased by more than 2% NS and 5% MK.

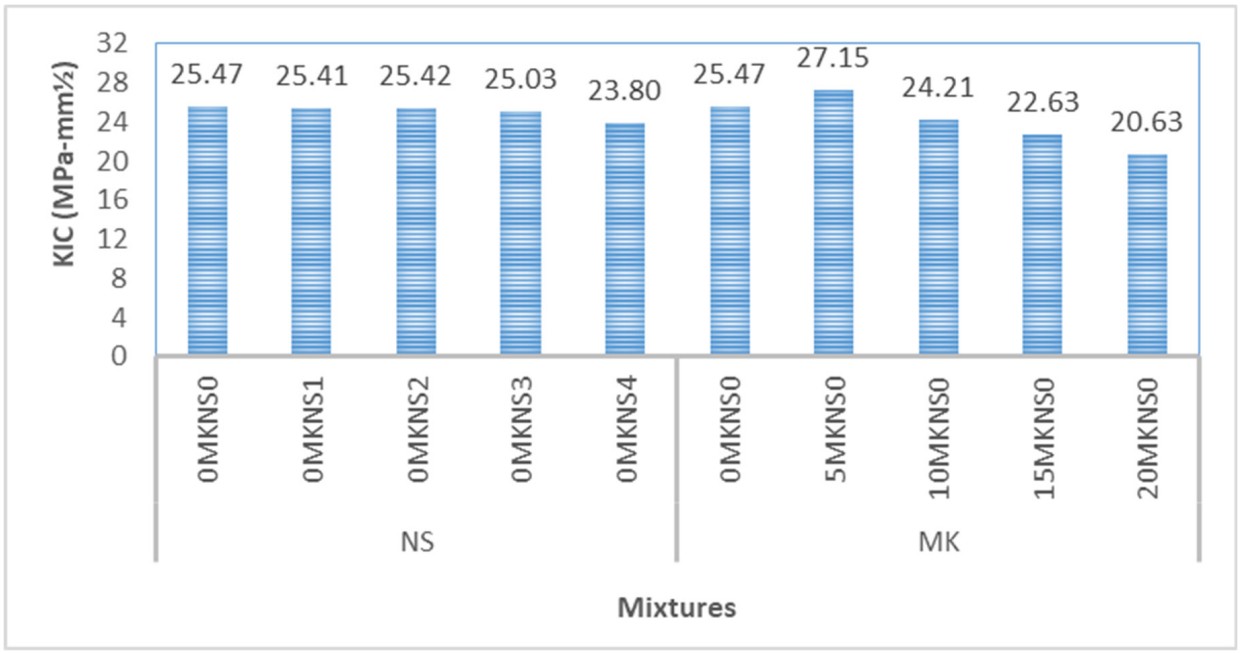

**Figure 14.** The stress intensity factor (KIC) of A-ASCC vs. MK and/or NS replacement ratios.

One of the study's goals for structural design codes and specifications of GGBS-based A-ASCC was to see if the current formula suggested for conventional concrete could be used for alkali-activated concrete. For this purpose, the Bazant and Becq-Giraduon [80] and CEB-FIP [81] proposed a conventional concrete formula (5th and 6th) relating compressive strength to fracture energy. These formulas were investigated and used to compare with the current study's fracture energy results.

$$G_f = 4.575 * \left( \frac{f_{c'}}{0.051} \right)^{0.46} \tag{5}$$

$$G_f = 25.69 * \left( \frac{f_{c'}}{10} \right)^{0.7} \tag{6}$$

As established, the suggested formulae matched the experimental outcomes, with an average error of 2.35% for the Bazant and Becq-Giraduon models, and 5.38% for the CEB-FIP model (Figure 15). Furthermore, the experimental results and suggested formulae matched up to 68 MPa. As a result, these formulae may be used to design the structural components of standard and high-strength FA-based A-ACC.

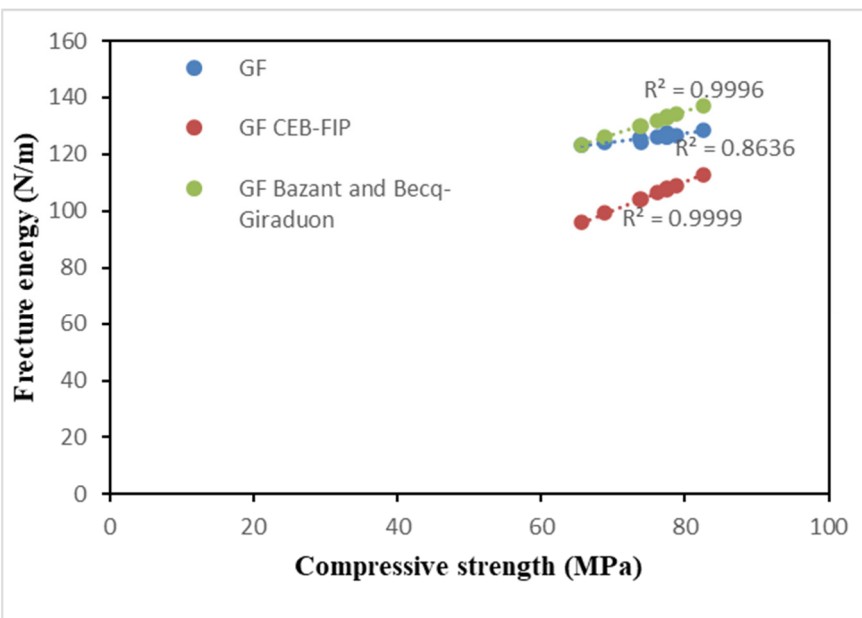

**Figure 15.** The fracture energy vs. compressive strength relationship.

### 3.3. Relationship and Correlation among the Properties of A-ASCC

Researchers must determine the correlation of experimental data to assess the experimental outcomes. In theory, the key elements influencing the fresh and mechanical properties of concretes are aggregate, w/b, and cementitious binder content. As previously specified, the compressive strength values of concrete have a substantial effect on the mechanical properties of the concrete. The effect of MK and NS incorporation on the characteristics of A-ASCC was investigated in this study. As a result, the fresh and hardened characteristics of A-ASCC based on this parameter, as well as the correlation and relationship between the experimental data, were investigated. As presented in Figure 16, close relationships were found between V-funnel and slump flow ($R^2$: 0.9181); L-box and slump flow ($R^2$: 0.9491); slump flow and T50 time ($R^2$: 0.9939); V-funnel and T50 ($R^2$: 0.9368); L-box and T50 ($R^2$: 0.9776); and V-funnel vs. L-box ($R^2$: 0.9518). Based on the $R^2$ values, it is possible to conclude that there is a strong relationship between slump flow and other properties, even in the presence of both MK and NS.

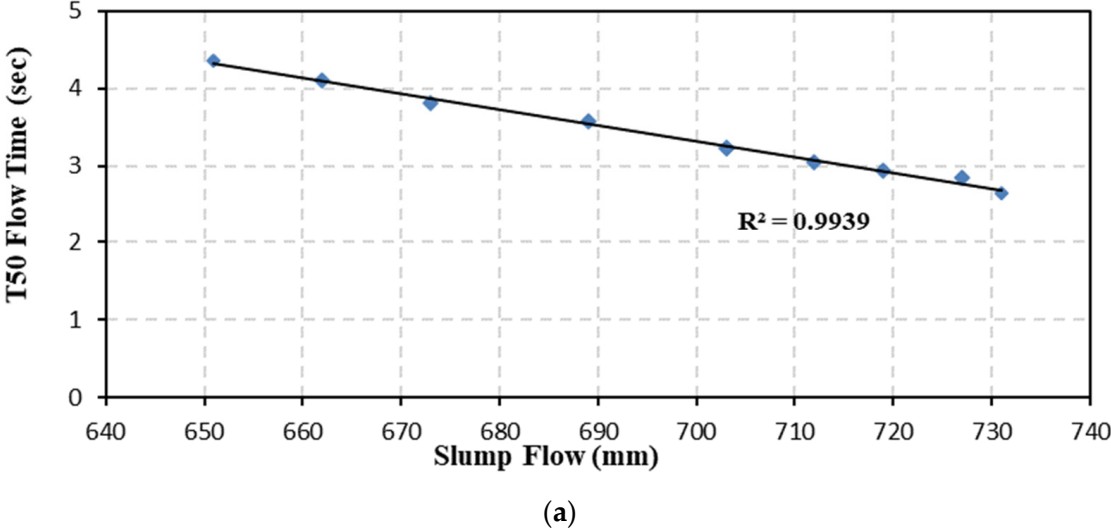

(**a**)

**Figure 16.** *Cont.*

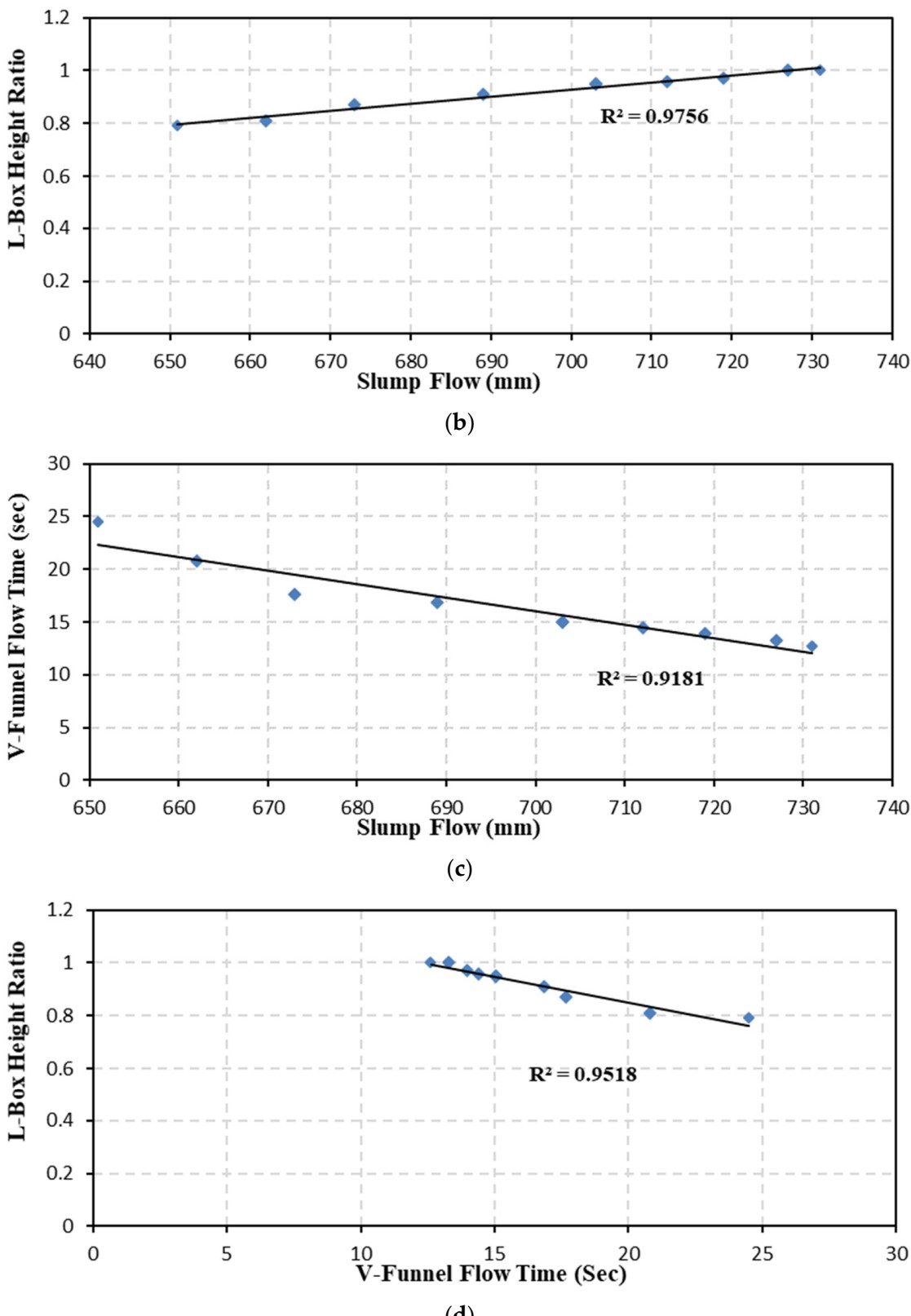

**Figure 16.** *Cont.*

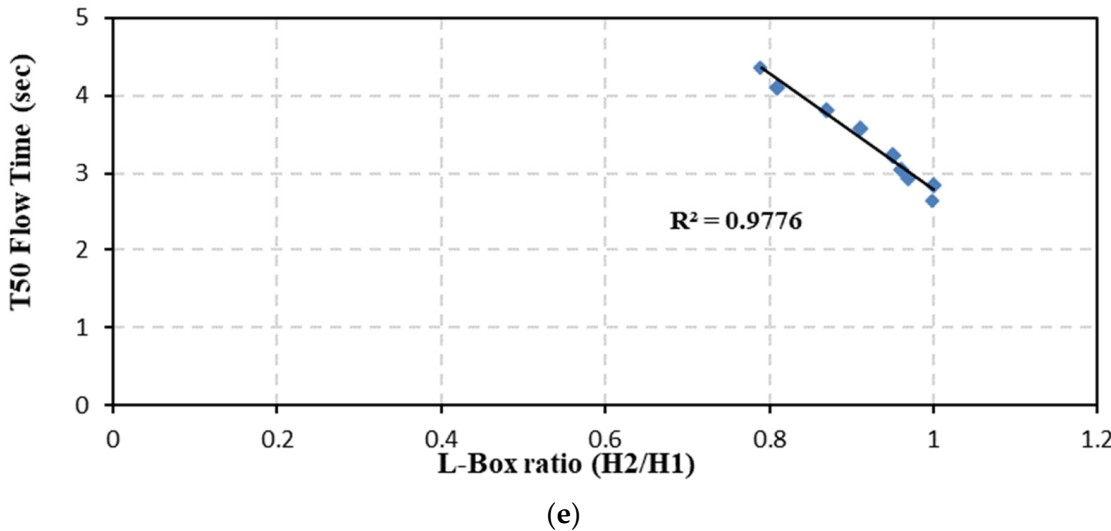

(**e**)

**Figure 16.** Correlation between the fresh properties of A-ASCC. (**a**) T50 vs. slump flow; (**b**) L-box vs. slump flow, (**c**) V-funnel vs. slump flow; (**d**) V-funnel vs. L-box; (**e**) L-box vs. T50.

The results, on the other hand, showed a notable strong correlation between the mechanical characteristics and slump flow of A-ASCC, as seen in Figure 17. The excellent correlation shows high coefficient ($R^2$) values, indicating that slump flow values had a considerable influence on the mechanical properties of A-ASCC, despite the existence of MK and NS. Furthermore, there are good relationships between the hardened performance of A-ASCC and the incorporation of MK and NS. As illustrated in Figure 18, the correlation between compressive strength and net flexural tensile strength, compressive strength and bond strength, and compressive strength and fracture energy is strong. Despite the addition of MK and NS, it is possible to conclude that there are superior correlations between A-ASCC hardened performance and fresh performance. The fresh properties of A-ASCC had a substantial effect on its hardened properties. As a result, good fresh characteristics are an essential requirement to achieve superior A-ASCC specimens with good mechanical properties.

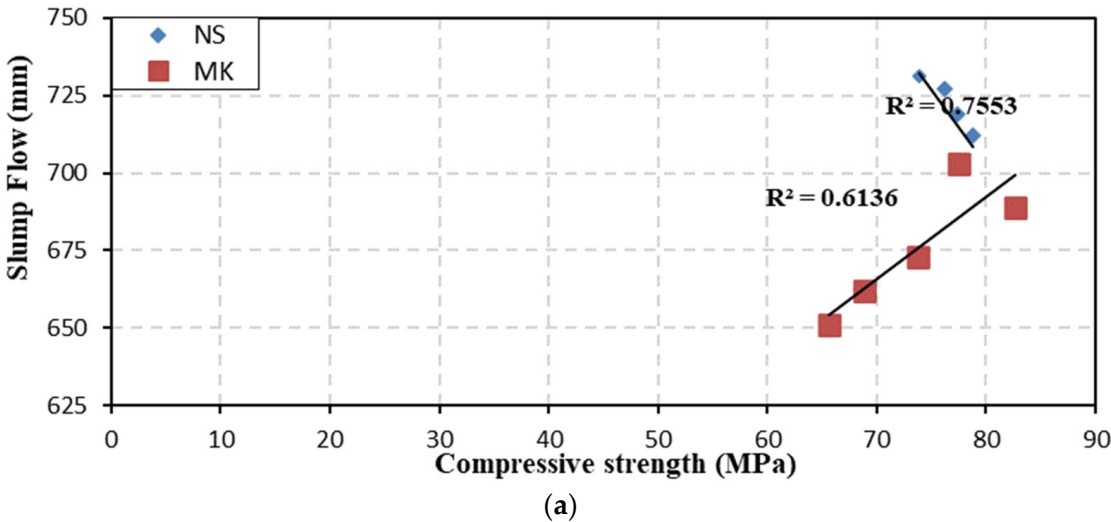

(**a**)

**Figure 17.** *Cont*.

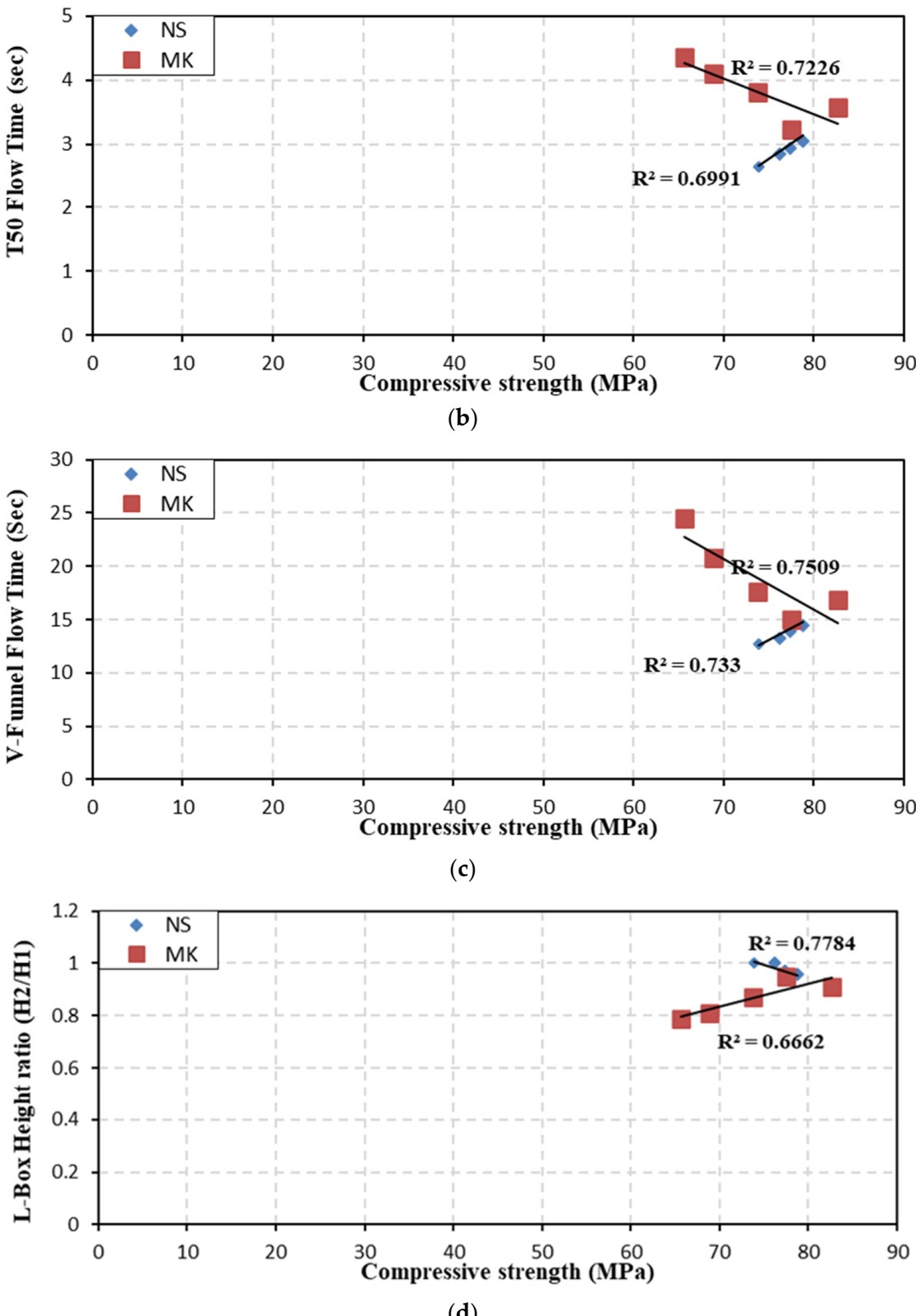

**Figure 17.** The hardened vs. fresh properties of the A-ASCC relationship. (**a**) Compressive strength vs. slump flow; (**b**) compressive strength vs. T50 flow time; (**c**) compressive strength vs. V-funnel flow time; (**d**) compressive strength vs. L-box passing ability.

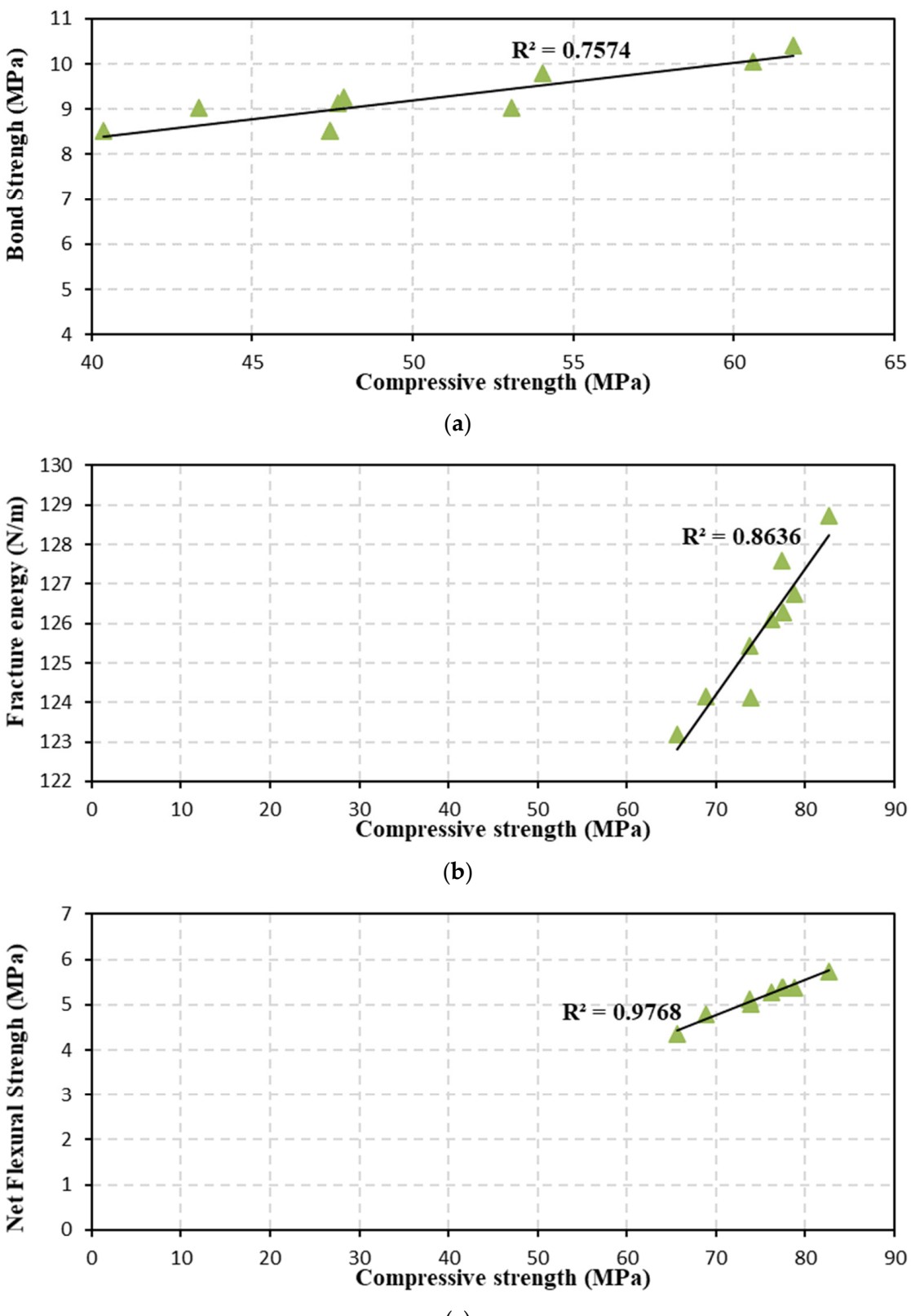

**Figure 18.** *Cont.*

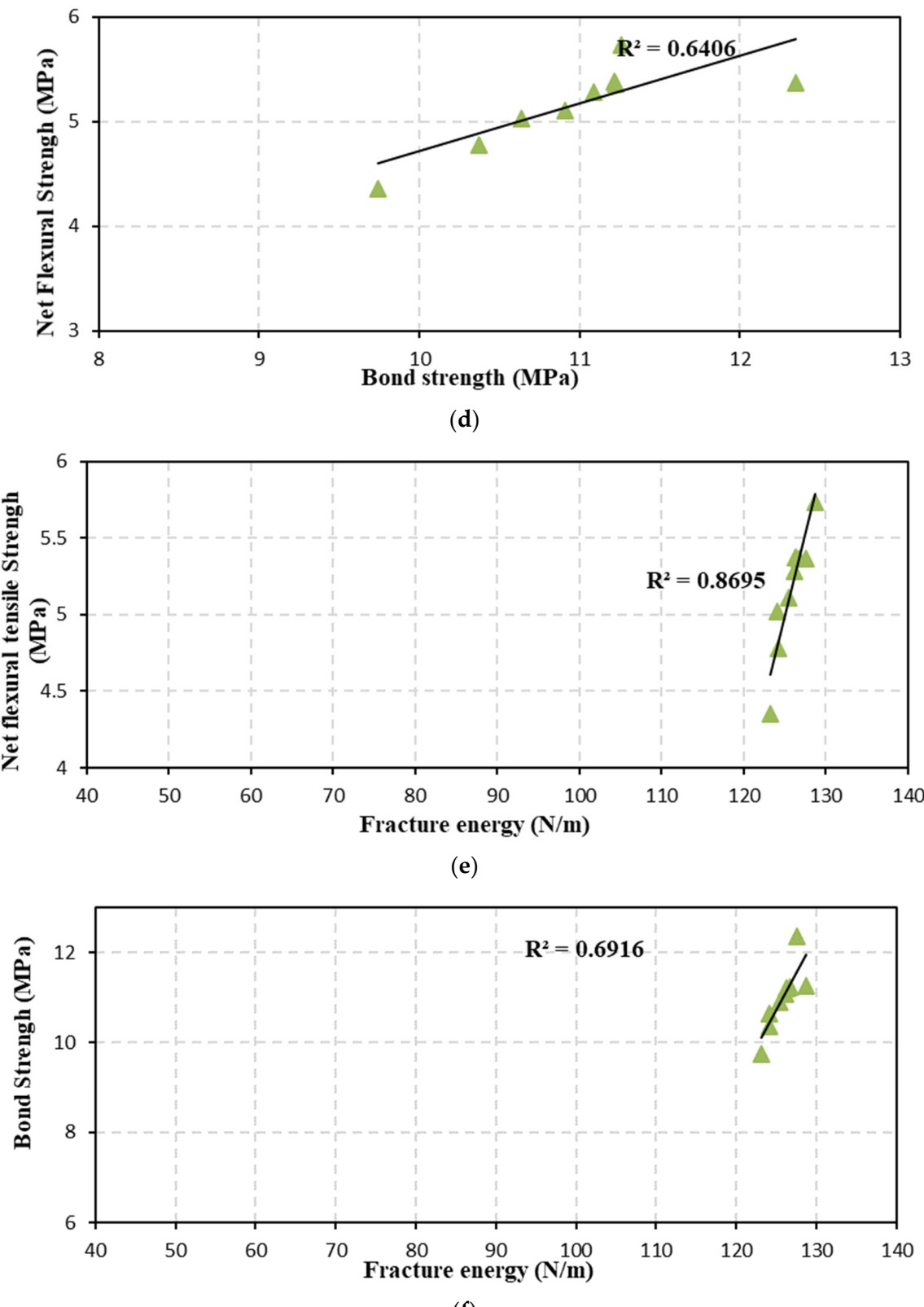

**Figure 18.** *Cont.*

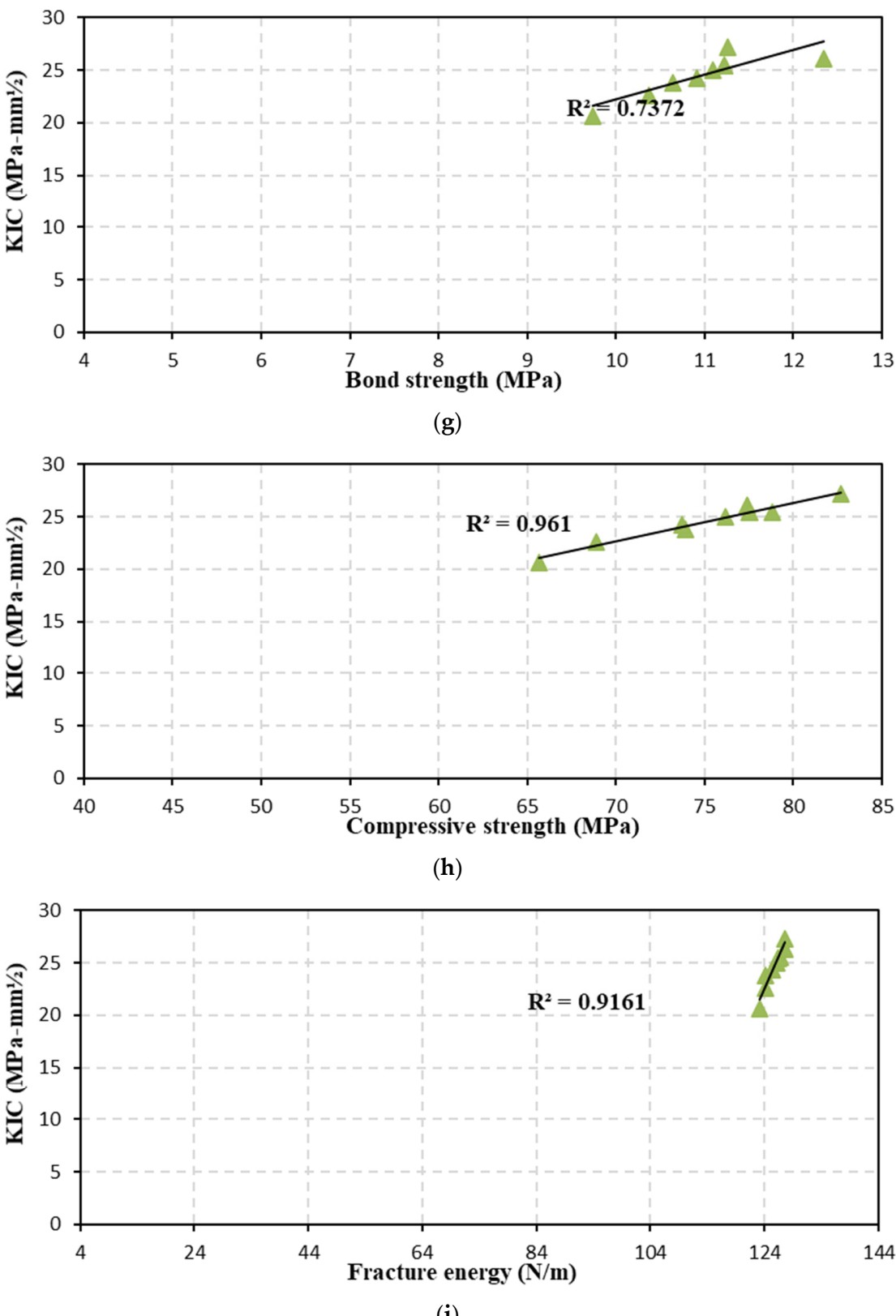

**Figure 18.** *Cont.*

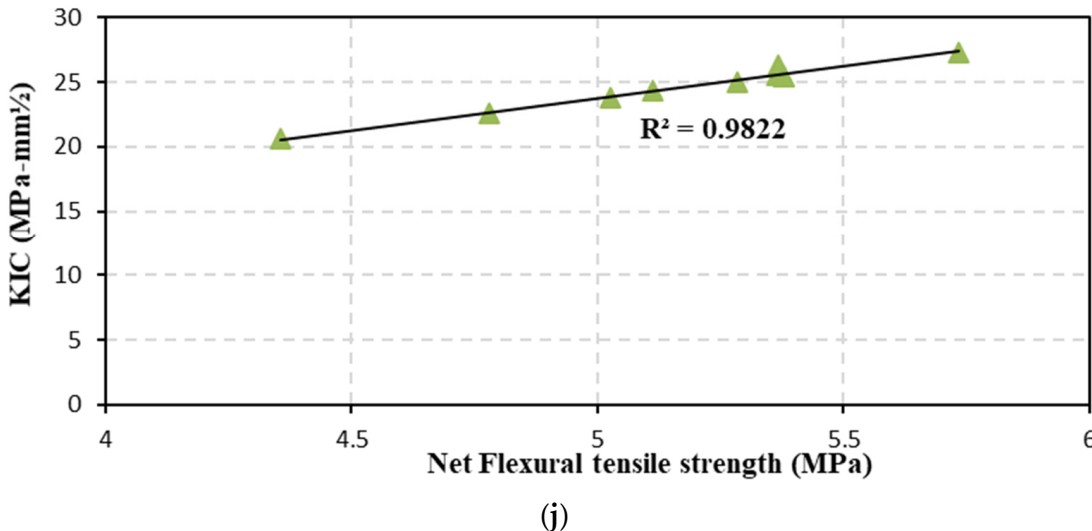

(**j**)

**Figure 18.** The hardened performance of A-ASCC relationships. (**a**) Bond strength vs. compressive strength; (**b**) fracture energy vs. compressive strength; (**c**) compressive strength vs. net flexural tensile strength, (**d**) net flexural tensile strength vs. bond strength; (**e**) net flexural tensile strength vs. fracture energy; (**f**) fracture energy vs. bond strength; (**g**) bond strength vs. KIC; (**h**) compressive strength vs. KIC; (**i**) fracture energy vs. KIC; (**j**) net flexural tensile strength vs. KIC.

*3.4. Statistical Analysis of the Experimental Result*

To examine the variation in the performance of the A-ASCC with varied concentrations of NS and/or MK in a detailed form, general linear model analysis of variance (GLM-ANOVA) was conducted at a significant level of 0.05. L-box height ratio, slump flow time, V-funnel flow time, slump flow diameter, fracture toughness, bond strength, net flexural tensile strength, compressive strength, and critical stress intensity factor were the dependent variables of the concretes, while MK and NS replacement levels were the independent variables. Statistical analysis was carried out to identify the statistically significant components (*p*-level 0.05). A *p*-value of less than 0.05 indicated that the relevant parameter played a significant role in the test outcomes. Furthermore, the % contribution was determined to provide information on the parameter's impact on overall performance. If this number is bigger, it is acceptable, since the parameter has a significant impact on overall performance. Table 7 shows the percentage contributions of each component to the final findings. The MK substitution was shown to have a greater influence on fresh state findings than the NS addition (V-funnel flow time, L-box ratio, T500 flow time, slump flow). Furthermore, the influence of MK on the compressive strength of the specimens was shown to be greater than that of NS. The introduction of NS and MK, on the other hand, was determined to have the greatest influence on bond and flexural performance.

A similar result was observed in the previous study. The fracture energy, bending strength, and bond/or bond strength of A-ASCC with and without NS and steel fiber was investigated by previous studies; the NS significantly improved the bending and bond strength of A-ASCC, and the enhancement increased with the combined use of NS and steel fiber [12,22]. Guneyisi et al. [69] investigated the impact of MK and steel fiber on the properties of conventional concrete and determined that the type and volume of steel fiber had a slight effect on the compressive strength of concrete; however, the effect of steel fiber was shown to be dominant on the bonding strength and flexural tensile strength. The combination of these two elements together reduces fresh state performance. Meanwhile, by mixing steel fiber and MK, the mechanical properties of the specimens were substantially enhanced.

**Table 7.** Statistical analysis of the study results.

| Dependent Variable | Independent Variable | Sequential Sum of Squares | Mean Square | Computed F | *p*-Value | Significant | Contribution (%) |
|---|---|---|---|---|---|---|---|
| Compressive Strength | MK replacement level | 140.4 | 140.4 | 0.66 | 00 | Yes | 88.98 |
| | NS replacement level | 9.8 | 9.8 | 10.0 | 0.00 | Yes | 6.21 |
| | Error | 7.58 | | | | | 4.8 |
| | Total | 157.78 | | | | | |
| Bond Strength | MK replacement level | 1.48 | 1.48 | 25.21 | 0.001 | Yes | 57.4 |
| | NS replacement level | 0.17 | 0.17 | 0.35 | 0.000 | Yes | 6.54 |
| | Error | 0.93 | | | | | 36 |
| | Total | 0.99980 | | | | | |
| Fracture Toughness | MK replacement level | 9.3 | 9.3 | 9.9 | 0.001 | Yes | 67.14 |
| | NS replacement level | 2.41 | 2.41 | 1.77 | 0.005 | Yes | 17.42 |
| | Error | 2.36 | | | | | 15.435 |
| | Total | 13.84 | | | | | |
| KIC | MK replacement level | 20.16 | 20.16 | 11.64 | 0.001 | Yes | 85.52 |
| | NS replacement level | 1.38 | 1.38 | 2.72 | 0.000 | Yes | 5.86 |
| | Error | 2 | | | | | 8.6 |
| | Total | 23.57 | | | | | |
| Net Flexure Tensile Strength | MK replacement level | 0.9 | 0.9 | 11.82 | 0.001 | Yes | 64.77 |
| | NS replacement level | 0.65 | 0.65 | 2.9 | 0.002 | Yes | 4.7 |
| | Error | 0.27 | | | | | 30.53 |
| | Total | 0.42 | | | | | |
| Displacement | MK replacement level | 0.004 | 0.004 | 42.85 | 0.000 | Yes | 13 |
| | NS replacement level | 0.006 | 0.006 | 53.57 | 0.00 | Yes | 20.35 |
| | Error | 0.02 | | | | | 66.62 |
| | Total | 0.03 | | | | | |

**Table 7.** *Cont.*

| Dependent Variable | Independent Variable | Sequential Sum of Squares | Mean Square | Computed F | *p*-Value | Significant | Contribution (%) |
|---|---|---|---|---|---|---|---|
| Slump Flow | MK replacement level | 409.6 | 409.6 | 52.96 | 0.000 | Yes | 44.61 |
| | NS replacement level | 504.1 | 504.1 | 213 | 000 | Yes | 54.9 |
| | Error | 4.3 | | | | | 0.47 |
| | Total | 918 | | | | | |
| L-Box Height Ratio | MK replacement level | 0.106 | 0.106 | 52.78 | 00 | Yes | 28.59 |
| | NS replacement level | 0.185 | 0.185 | 149.1 | 00 | Yes | 49.85 |
| | Error | 0.08 | | | | | 21.57 |
| | Total | 0.37 | | | | | |
| V-Funnel Flow Time | MK replacement level | 2.08 | 2.08 | 45.4 | 00 | Yes | 35.4 |
| | NS replacement level | 3.52 | 3.52 | 792.3 | 00 | Yes | 59.82 |
| | Error | 0.28 | | | | | 4.76 |
| | Total | 5.89 | | | | | |
| T-50 Flow Time | MK replacement level | 0.001 | 0.001 | 15.42 | 00 | Yes | 7.07 |
| | NS replacement level | 0.002 | 0.002 | 36.75 | 00 | Yes | 9.62 |
| | Error | 0.0017 | | | | | 83.3 |
| | Total | 0.02 | | | | | |

## 4. Conclusions

The effect of MK and NS use on the fresh and hardened behavior of the GGBS-based A-ASCC was examined in this work. The findings are summarized below:

- The addition of nanosilica increased the V-funnel flow time, L-box passage ability, T50 flow time, and slump flow in fresh state tests. The mixtures with the highest NS showed the largest improvement in fresh performance (4%). The addition of MK, on the other hand, reduced the fresh properties of A-ASCC; the minimum fresh properties were deducted by using 20% MK. Even in this case, the A-ASCC mixes satisfied the EF-NARC and TS 12350 flowability and passing ability standards.
- Adding additional NS and/or MK to the A-ASCC mixture improved the resistance to segregation and bleeding, whereas NS and/or MK-containing mixtures were shown to be more cohesive than non-NS and/or non-MK-containing mixes.
- Based on V-funnel and slump flow tests, EFNARC standards confirmed that all A-ASCC mixes were in the VS2/VF2 viscosity class, which has superior bleeding and segregation resistance and low formwork pressure.

- The addition of nanosilica (up to 2%), and MK (up to 5%), marginally improved the mechanical properties of A-ASCC, which were decreased with an increase in the NS ratio (more than 2%), and the MK ratio (more than 5%).
- The addition of NS and/or MK considerably increased the bond strength, flexural strength, fracture energy, and stress intensity factor of A-ASCC specimens. The specimens with concentrations of metakaolin (5%) and/or nanosilica (up to 2%) exhibited the best mechanical performance.
- The maximum bond strength was obtained by using NS and/or MK. The maximum improvement was 9.62% and 0.35% for specimens with 2% NS and 5% MK, respectively. As a result, it can be stated that the optimal NS and/or MK ratio for achieving the best fresh and hardened performance of GGBS-based A-ASCC was 2% and 5%, respectively.
- The data analysis showed that all of the independent variables had a significant influence on the properties of A-ASCC (both fresh and hardened properties). The most critical components were shown to be the addition of nanosilica and metakaolin. The presence of metakaolin had a stronger impact on the specimens' fresh performance (V-funnel flow time, L-box passage ability, T50 flow time, and slump flow) and compressive strength than the addition of nanosilica. The incorporation of NS and MK, on the other hand, was found to be the most important variable in the A-ASCC specimens' flexural tensile strength, fracture energy, and bond strength performance.

**Funding:** This research received no external funding.

**Institutional Review Board Statement:** Not applicable.

**Informed Consent Statement:** Not applicable.

**Data Availability Statement:** Generated during the experimental study.

**Acknowledgments:** Kindly: I would like to express thanks and my sincere gratitude to Abdulkadir Çevik, and the University of Gaziantep, department of civil engineering for continuous support during the experimental work and for providing guidance in the present for a successful academic future.

**Conflicts of Interest:** The authors declare no conflict of interest.

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
