# Peer review of "Bond Strength and Fracture Toughness of Alkali Activated Self-Compacting Concrete Incorporating Metakaolin or Nanosilica"

_sustainability, doi:10.3390/su14116798_

Round 1
Reviewer 1 Report
1.Rewrite "However, the influence of NS on the compressive strength of A- 24
ASCC was unsignificant. The mechanical properties of A-ASCC were improved by 5% replacement 25
ratio of metakaolin replacement ratios. The use of metakaolin more than 5% exhibits negative effect 26
on the properties of A-ASCC. The greater improvement was shown on the A-ASCC bond strength 27
for both NS and MK. In addition, the use of NS and/or MK significantly increased the A-ASCC 28
setting time. As a result, it is possible to conclude that the A-ASCC may be used in an ambient 29
environment
2.Shorten "Although there have been various studies on SCC, there have been few or limited 134
studies on the hardened and fresh characteristics of A-ASCC. As a result, the aims of this 135
research are to clarify the effect of nanosilica and metakaolin on the fresh, bond strength 136
and mechanical properties of slag-based A-ASCC. As a result, two series of A-ASCC mix- 137
tures were developed: with metakaolin but no nanosilica and with nanosilica but no me- 138
takaolin. Slag was replaced with NS by weight at a rate of 0%, 1%, 2%, 3%, and 4%, re- 139
spectively; and replaced with MK by weight at a rate of 0%, 5%, 10%, 15%, and 20%, re- 140
spectively. Nine A-ASCC mixes were produced, each with a total binder content of 500 141
kg/m3 and a constant alkaline/binder ratio of 0.50. The workability and flow ability of A- 142
ASCC mixes were evaluated through the L-box ratio, V-funnel, slump flow, and T50 time 143
tests. The compressive strength, flexural tensile strength, bonding strength, and fracture 144
toughness of A-ASCC mixtures were determined while investigating their mechanical 145
properties. All experimental test results were statistically evaluated, and a correlation con- 146
nection analysis was shown to establish the significant effect of nanosilica and/or me- 147
takaolin on A-ASCC characteristics."
3.Discuss more "Table 1 illustrates the chemical 154
compositions and physical properties of the NS, MK, and FA. To evaluate the effect of MK 155
and NS on the hardened and fresh performances of A-ASCC mixes, nine mixes were pro- 156
duced in two series, one with nanosilica (NS) (0, 1, 2, 3, and 4%), and the other with me- 157
takaolin (MK) (0, 5, 10, 15, and 20%), and one without NS and MK as reference. Slag (an 158
industrial waste mineral), was utilized as a binder material in the current study. The slag 159
weight was used to replace the NS and MK. "
4.How to control the mix proportions "Table 3 shows the quantity of each component of 177
A-ASCCs mixes (weight/m3 concrete). For mixture designations, NS represents nanosil- 178
ica, the number next to NS (0-4) represents the nanosilica replacement ratio, and the num- 179
ber in front of MK (0-5-10-15-20) represents the quantity of MK utilized in the production 180
of A-ASCCs mixes. "?
5.Explain more " The ultimate setting times 289
for these mixes were 4 hrs., 4 hrs., 3 hrs. and 2 hrs., respectively. "
6.Recheck and rewrite "In the 483
current study the bond strength of A-ASCC was slightly improved with the replacement 484
of MK and NS as shown in Figure 12. However, the improvement achieved by the MK 485
and NS replacement ratio was close to each other. "
Author Response
Dear Professor:
Thank you indeed for your valuable comments. the response to your comments is attached below.
Regards...

Reviewer 2 Report
Author has presented an interesting study on “Bond Strength behavior and Fracture Toughness of Slag Based Alkali Activated Self-Compacting Concrete inclusion Metakaolin and/or Nanosilica”. This paper showed an extensive experimental work. How author justify its sole authorship ignoring others contribution? Article needs serious improvement in terms of its writing style and language. Repetitions observed throughout the manuscript which should be avoided. Please write as a scientific article rather than a lab report.
Other than above, below are some additional comments and suggestions for authors to improve this manuscript.
(1) Title is too long and needs modification. For inclusion author may use incorporating.
(2) Abstract is too long and needs to highlight only importance of this work and main outcomes.
(3) The introduction part should focus only on specific information relevant to current research. Most of its initial part is fundamental and known facts.
(4) Please be consistent in writing as slag or GGBFS as there are different types of slag in the market
(5) In Table 3, what's the difference between mix-1 and mix-6? Why repeated?
(6) What you mean by extra water in Table 3? Why dosages of SP were kept constant?
(7) What kind of mixing device was used? How did you consider dispersion effect of NS.
(8) Fig. 3a and 3c should be deleted. Fig. 3b is enough and show only a clear photo.
(9) Please merge Fig. 4 and Fig. 5b and delete Fig. 5a.
(10) Improve graphical presentation of results. Current presentation of graphs is so poor to put in scientific article.
(11) Please revise conclusion and write only major findings.
Author Response

(The authors gave the same response as above.)

Reviewer 3 Report
My comments to enhance the manuscript quality are following:
- Abstract (Line 14), please replace slag by GBFS.
- Most chemical formula of sodium silicate wrongly written.
- Abstract too long and authors should focus on the results achieved aims of this study.
- Introduction section: first paragraph just general information and author can remove it and focus on benefits of MK and NS for bond strength behavior of alkali-activated concrete.
- Introduction section, second paragraph not shown what the problem of self-compacting concrete prepared with GBFS only.
- Gap and novelty of this not clear.
- Please highlight the aims of this research.
- Fig. 1 not important and can remove it.
- In manuscript title (MK and/or NS), please remove and as in mix design no (MK and NS). see Table 3.
- Fig. 3 not added any value to manuscript content.
- Line 279-281, what the meaning?
- line 285, change 125 hours to 1.25 hrs.
- What the effect of chemical composition of GBFS, NS and MK on flowability and setting time of proposed concrete?
- Results presented in section 3.1, not well discussed.
- Please highlight sustainable benefits from inclusion MK and NS in concrete mix.
- Manuscript not well organized.
Author Response

(The authors gave the same response as above.)

Round 2
Reviewer 1 Report
Authors have well improved their manuscript.
Reviewer 2 Report
Authors have well improved their manuscript.
Reviewer 3 Report
Author modified the manuscript content.